# Thermal Endurance by a Hot-Spring-Dwelling Phylogenetic Relative of the Mesophilic *Paracoccus*

Nibendu Mondal,[a] Chayan Roy,[a*] Sumit Chatterjee,[a] Jagannath Sarkar,[a] Subhajit Dutta,[a] Sabyasachi Bhattacharya,[a] Ranadhir Chakraborty,[b] Wriddhiman Ghosh[a]

aDepartment of Microbiology, Bose Institute, Kolkata, India
bDepartment of Biotechnology, University of North Bengal, Siliguri, India

**ABSTRACT** High temperature growth/survival was revealed in a phylogenetic relative (SMMA_5) of the mesophilic *Paracoccus* isolated from the 78 to 85°C water of a Trans-Himalayan sulfur-borax spring. After 12 h at 50°C, or 45 min at 70°C, in mineral salts thiosulfate (MST) medium, SMMA_5 retained ~2% colony forming units (CFUs), whereas comparator *Paracoccus* had 1.5% and 0% CFU left at 50°C and 70°C, respectively. After 12 h at 50°C, the thermally conditioned sibling SMMA_5_TC exhibited an ~1.5 time increase in CFU count; after 45 min at 70°C, SMMA_5_TC had 7% of the initial CFU count. 1,000-times diluted Reasoner's 2A medium, and MST supplemented with lithium, boron, or glycine-betaine, supported higher CFU-retention/CFU-growth than MST. Furthermore, with or without lithium/boron/glycine-betaine, a higher percentage of cells always remained metabolically active, compared with what percentage formed single colonies. SMMA_5, compared with other *Paracoccus*, contained 335 unique genes: of these, 186 encoded hypothetical proteins, and 83 belonged to orthology groups, which again corresponded mostly to DNA replication/recombination/repair, transcription, secondary metabolism, and inorganic ion transport/metabolism. The SMMA_5 genome was relatively enriched in cell wall/membrane/envelope biogenesis, and amino acid metabolism. SMMA_5 and SMMA_5_TC mutually possessed 43 nucleotide polymorphisms, of which 18 were in protein-coding genes with 13 nonsynonymous and seven radical amino acid replacements. Such biochemical and biophysical mechanisms could be involved in thermal stress mitigation which streamline the cells' energy and resources toward system-maintenance and macromolecule-stabilization, thereby relinquishing cell-division for cell-viability. Thermal conditioning apparently helped inherit those potential metabolic states which are crucial for cell-system maintenance, while environmental solutes augmented the indigenous stability-conferring mechanisms.

**IMPORTANCE** For a holistic understanding of microbial life's high-temperature adaptation, it is imperative to explore the biology of the phylogenetic relatives of mesophilic bacteria which get stochastically introduced to geographically and geologically diverse hot spring systems by local geodynamic forces. Here, *in vitro* endurance of high heat up to the extent of growth under special (habitat-inspired) conditions was discovered in a hot-spring-dwelling phylogenetic relative of the mesophilic *Paracoccus* species. Thermal conditioning, extreme oligotrophy, metabolic deceleration, presence of certain habitat-specific inorganic/organic solutes, and potential genomic specializations were found to be the major enablers of this conditional (acquired) thermophilicity. Feasibility of such phenomena across the taxonomic spectrum can well be paradigm changing for the established scopes of microbial adaptation to the physicochemical extremes. Applications of conditional thermophilicity in microbial process biotechnology may be far reaching and multifaceted.

**KEYWORDS** hot spring mesophiles, *Paracoccus*, thermal endurance, thermal conditioning, acquired thermophilicity, genome specialization, oligotrophy, boron, lithium, glycine-betaine

Address correspondence to Wriddhiman Ghosh, wriman@jcbose.ac.in, or Wriman@rediffmail.com.

*Present address: Chayan Roy, Department of Plant and Environmental Sciences, University of Copenhagen, Copenhagen, Denmark.

The authors declare no conflict of interest.

Association of the typically thermophilic or hyperthermophilic bacteria and archaea with terrestrial hydrothermal habitats is axiomatic. Accordingly, our knowledge on microbial adaptation to high temperature is based largely on hot spring isolates that grow *in vitro* either obligately at ≥80°C (1, 2) or facultatively between 30°C and 80°C (3, 4). Members of mesophilic microbial groups (taxa having no member reported for laboratory growth at >45°C), on the other hand, though unexpected in high-temperature environments, often get stochastically introduced by local geodynamic forces to the hot spring systems (5–7), where they are detected mostly via sequencing and analysis of metagenomes (6–15), and sometimes as pure culture isolates (12, 16, 17). In such a scenario, for a holistic understanding of microbial life's high-temperature adaptation, it becomes imperative to explore the biology of the phylogenetic relatives of mesophilic bacteria which happen to be there in geographically and geologically distinct hot spring habitats. That said, most studies on thermotolerance by mesophilic bacteria have thus far been centered on economically or clinically important strains isolated from environments other than hydrothermal ecosystems (18–22). Here, we investigate high-temperature growth/survival in relation to the physiology, cell biology, and genomics of a novel, facultatively chemolithoautotrophic strain of the alphaproteobacterial genus *Paracoccus* (named SMMA_5) that was isolated on mineral salts thiosulfate (MST) medium (12) from the vent-water of a Trans-Himalayan sulfur-borax spring called Lotus Pond, located in the Puga geothermal area of eastern Ladakh (India), at an altitude of 4,436 m, where water boils at ~85°C (23). Remarkably, whereas no member of *Paracoccus* grows at >45°C *in vitro*, the temperature of the habitat of SMMA_5 ranges between 78°C and 85°C (7, 12, 23).

To understand how SMMA_5 responds to thermal stress, increase/decrease in the number of colony forming units (CFUs) present in the culture was tested at different temperatures (37°C to 70°C); frequency of viable (metabolically active) cells within the culture was determined via fluorescein diacetate (FDA) staining followed by flow cytometry. Potential effects of thermal conditioning on the isolate's ability to grow or survive at high temperatures were tested by subjecting the organism to different iterations of "heat exposure and withdrawal" cycles.

Previous geomicrobiological explorations of Lotus Pond had shown that the snow-melts and denuded soil/sediments, which infiltrate the shallow geothermal reservoir of Puga via local tectonic faults, introduce mesophilic bacteria into the hot spring system (7). Since the soil/sediment systems of the frigid deserts of Ladakh are typically poor in organic carbon (24), and because the vent-water of Lotus Pond has low dissolved solutes concentration (7), the plausible role of oligotrophy in thermal endurance by SMMA_5 was tested via high-temperature incubation in different dilution grades of Reasoner's 2A (R2A) medium, which is universally used for growing oligotrophic bacteria (25).

At equivalent temperature levels, hot spring waters having neutral to moderately alkaline pH generally harbor greater microbial diversities than their acidic counterparts (7). In view of this, and in consideration of the circum-neutral pH (diurnal range: 7.2 to 8.0) of the Lotus Pond vent-water (7, 23), we tested the effects of pH 7.0 to 9.0 on the high-temperature growth/survival of SMMA_5.

Dissolved solutes such as ions of boron and lithium, which are typical of the Puga hot springs, including Lotus Pond (7, 14, 23), have been hypothesized previously as the *in situ* mitigators of the biomacromolecule-disordering effects of heat, and therefore the facilitators of the colonization of high-temperature sites by mesophilic bacteria (14). Accordingly, sodium tetraborate and lithium hydroxide were tested for their potential effects on the high-temperature growth/survival of SMMA_5. Furthermore, previous culture-independent community analyses of Lotus Pond's vent-water (7, 12) had revealed the presence of several species belonging to glycine-betaine-producing genera such as *Bacillus*, *Ectothiorhodospira*, *Escherichia*, *Halomonas*, *Pseudomonas*, and *Staphylococcus* (26–28). Since the osmoprotective compatible solute glycine-betaine is known to mitigate a wide range of physicochemical stressors (29), its potential effect

on the high-temperature growth/survival of SMMA_5 was tested alongside boron and lithium. Finally, to know whether thermal endurance had a genetic foundation, the whole-genome of SMMA_5 was sequenced and analyzed in conjunction with that of its thermally conditioned variant SMMA_5_TC.

## RESULTS

**Taxonomic identity and growth characteristics of SMMA_5.** The 16S rRNA gene sequence of SMMA_5 exhibited the closest (96.2 to 97.5%) similarities with homologs from diverse *Paracoccus* species; in the phylogenetic tree constructed, the isolate shared a major clade with *Paracoccus denitrificans*, *Paracoccus kondratievae*, *Paracoccus pantotrophus*, *Paracoccus versutus*, *Paracoccus bengalensis*, and a number of other species (see Fig. S1 in the supplemental material). In the phylogeny reconstructed based on 92 conserved marker gene sequences (Fig. 1), SMMA_5 clustered in a major clade encompassing *Paracoccus aestuariivivens*, *Paracoccus aminophilus*, *Paracoccus aminovorans*, *P. denitrificans*, *P. kondratievae*, *Paracoccus laeviglucosivorans*, *Paracoccus limosus*, *Paracoccus litorisediminis*, *Paracoccus lutimaris*, *P. pantotrophus*, *Paracoccus sulfuroxidans*, *Paracoccus thiocyanatus*, *P. versutus*, and *Paracoccus yeei* (notably, no genome sequence was available for *P. bengalensis*, so this organism could not be included in the phylogenomic analysis).

The SMMA_5 genome exhibited 23.7%, 23.8%, and 21.6% DNA-DNA hybridization, *in silico*, with the phylogenetically/phylogenomically closest species *P. aminovorans*, *P. denitrificans* and *P. kondratievae*, respectively. Orthologous gene-based average nucleotide identities of SMMA_5 with the above three genomes were 81.2%, 81.4%, and 78.8%, respectively. These data indicated that SMMA_5 was a potentially novel species of *Paracoccus*.

When SMMA_5 was incubated in MST medium for 12 h, $>10^4$ times increases in CFU count were observed at 37°C to 45°C, with a generation time of 91.4, 98.8, and 99.1 min recorded at 37°C, 40°C, and 45°C, respectively; at 50°C, 2% of the initial CFU count remained in the culture, whereas at 60°C and 70°C no CFU was left (Fig. 2A and B). However, until 45 min in MST at 70°C, SMMA_5 had 1.6% of the initial CFU count remaining in the culture (Fig. 3A and B).

The comparator chemolithoautotroph *P. pantotrophus* LMG 4218, isolated previously from a denitrifying, sulfide-oxidizing, effluent treatment plant (30), when incubated in MST for 12 h, exhibited $>10^4$ times increase in CFU count at up to 40°C; at 45°C, $\sim10^3$ times increase in CFU count was recorded, while at 50°C, 1.5% of the initial CFU count remained in the culture; at 60°C and 70°C no CFU was left (Fig. 2C and D); there was also no CFU after 45 min in MST at 70°C (Fig. 3A and B). Generation time of LMG 4218, over 12 h in MST at 37°C, 40°C, and 45°C, was 95.6, 99.2, and 213.7 min, respectively.

*Escherichia coli* K-12, isolated previously from the feces of a convalescent diphtheria patient (31), when incubated in mineral salts dextrose (MSD) for 12 h showed $>10^4$ times CFU growth at up to 40°C; whereas only 5.8% CFU remained in the culture at 45°C, no CFU was left at $\geq$50°C (Fig. 2E and F); there was also no CFU after 45 min at 70°C (Fig. 3A and B). Generation time of K-12, over 12 h in MSD at 37°C and 40°C, was 89.6 and 96.4 min, respectively.

**Thermal conditioning enhances high-temperature growth/survival.** For maintenance, SMMA_5 was grown for 12 h at 37°C and stored at 4°C for 28 days until the next transfer. On the other hand, its thermally conditioned cell line (sibling strain), designated SMMA_5_TC, was raised and maintained by first incubating for 12 h at 50°C, then reviving growth for 12 h at 37°C, and finally storing at 25°C for 14 days until the next transfer involving the same 50°C to 37°C incubation procedure (Fig. 4A). All data presented in this study for SMMA_5 and SMMA_5_TC were obtained from experiments carried out after at least 15 and 30 transfer cycles since strain establishment, respectively.

When tested for growth in MST medium at 37°C, 40°C, and 45°C, SMMA_5_TC (Fig. 2G and H) exhibited phenotypes identical to those of SMMA_5 (Fig. 2A and B). However, after 12 h at 50°C in MST, SMMA_5_TC exhibited $\sim$1.5 times increase in the

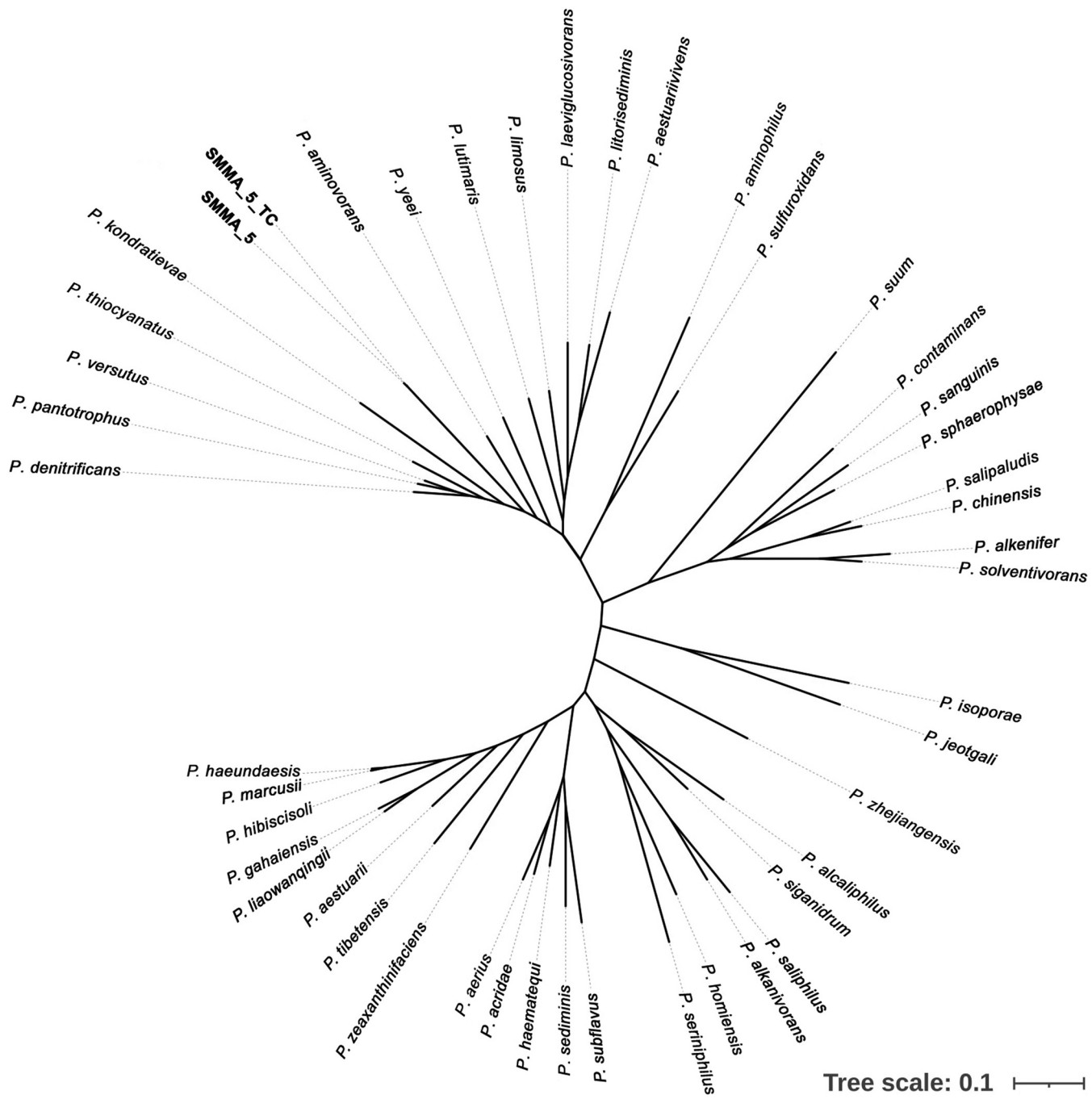

**FIG 1** Phylogenetic relationships delineated based on 92 conserved marker gene sequences for SMMA_5, SMMA_5_TC, and 44 other *Paracoccus* species which have nearly complete genome sequences available in GenBank (see Table S1).

CFU count of the culture with respect to the 0 h level (Fig. 2G and H), whereas under identical conditions SMMA_5 retained only 2% of the initial CFU count (Fig. 2A and B). Albeit, SMMA_5_TC (Fig. 2G and H) did not have any CFU left after 12 h in MST at 60°C and 70°C, after 45 min in MST at 70°C it had 7% of the initial CFU count intact (Fig. 3A and B).

Subsequently, experiments were carried out to know the effects of only a few thermal conditioning (heat exposure and withdrawal) cycles on the growth or CFU retention by SMMA_5 at 50°C and 70°C (each cycle involved incubation for 12 h at 50°C, followed by 12 h at 37°C; Fig. 4B). When the six sibling strains generated after one round of thermal conditioning were individually incubated for 12 h at 50°C, 2.5%, 4.3%, 1.7%,

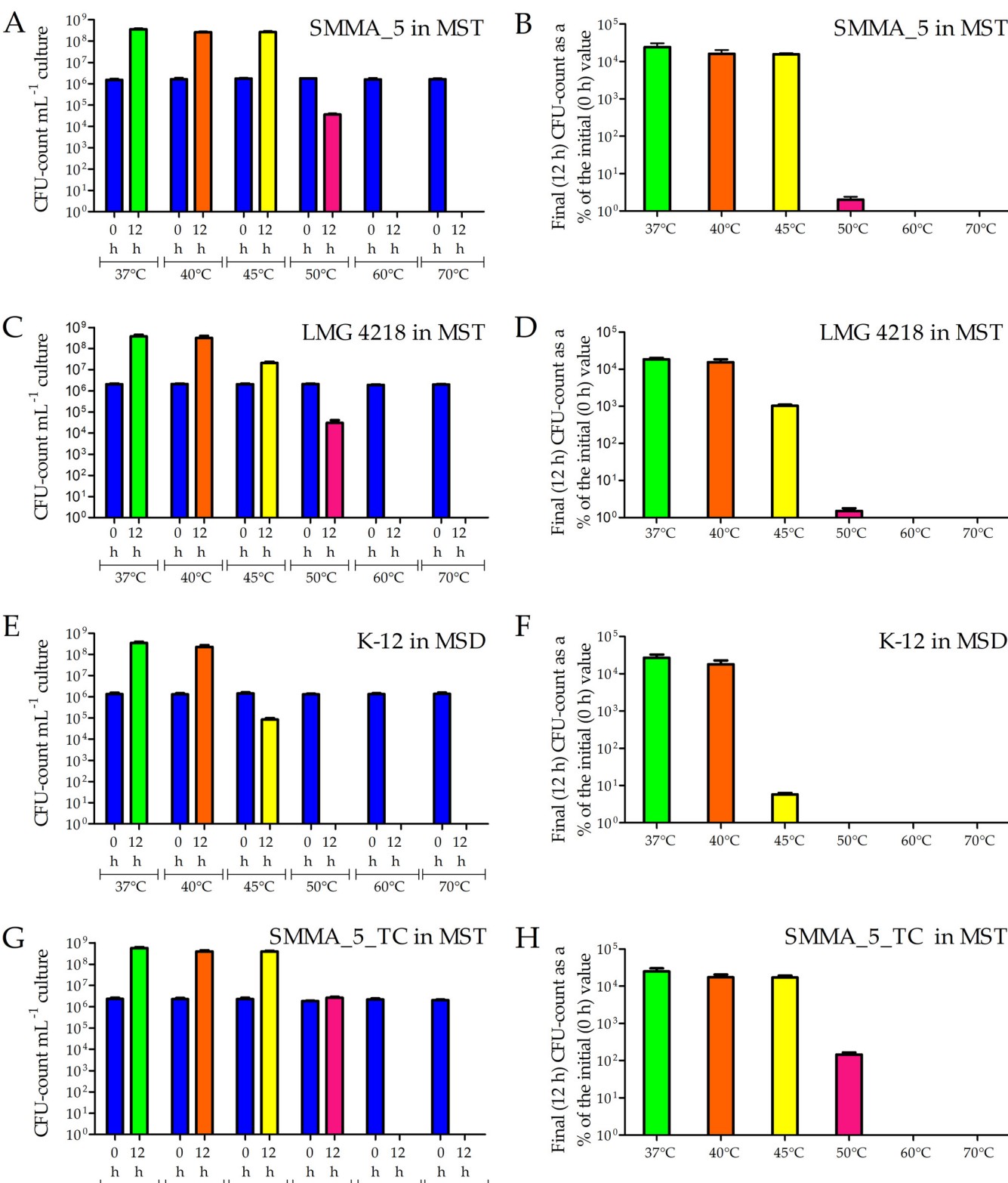

**FIG 2** Increase or decrease in the CFU counts of *Paracoccus* strains SMMA_5, LMG 4218, and SMMA_5_TC in modified basal and mineral salts solution supplemented with thiosulfate (MST), and *E. coli* strain K-12 in modified basal and mineral salts solution supplemented with dextrose (MSD), after 12 h of incubation at 37°C, 40°C, 45°C, 50°C, 60°C, and 70°C. (A) CFU counts for SMMA_5 at 0 h and 12 h recorded at the different incubation temperatures; (B) representing each final (12 h) CFU count datum of panel A as a percentage of the corresponding initial (0 h) CFU count. (C) CFU counts for LMG 4218 at 0 h and 12 h recorded at the different incubation temperatures; (D) each 12 h CFU count datum of panel C is represented as a percentage of the corresponding 0 h CFU count. (E) CFU counts for K-12 at 0 h and 12 h recorded at the different incubation temperatures; (F) each 12 h CFU count datum

3.7%, 5.8%, and 2.9% (on an average 3.5%) of their initial CFU counts were retained (Fig. 5A and B). Likewise, when these six sibling strains were individually incubated for 45 min at 70°C, 3.8%, 1.8%, 4.2%, 3.2%, 2.7%, and 4.0% (on an average 3.3%) of their initial CFU counts were retained (Fig. 5E and F).

When the six sibling strains generated after five rounds of thermal conditioning were individually incubated for 12 h at 50°C, 23.7%, 16.1%, 11.4%, 18.8%, 11.0%, and 8.9% (on an average 15%) of their initial CFU counts were retained (Fig. 5C and D). Furthermore, when these six sibling strains were individually incubated for 45 min at 70°C, 10.5%, 3.1%, 4.0%, 2.2%, 2.6%, and 15.2% (on an average 6.2%) of their initial CFU counts were retained (Fig. 5G and H).

**At high temperatures, the proportion of metabolically active cells in a culture exceeds the proportion of CFUs.** After 12 h at 50°C, as well as after 45 min at 70°C, the percentages of SMMA_5 and SMMA_5_TC cells remaining viable or metabolically active were far higher than the percentages of cells remaining ready to divide and form single colonies (CFU count data). As a mark of their metabolically active state, cell populations were tested post high-temperature incubation for their ability to take up FDA and hydrolyze it to fluorescein, which in turn was detected via flow cytometry. In that way, after 12 h of incubation at 50°C in MST, 98.7% and 99.56% of SMMA_5 and SMMA_5_TC cells were found to remain viable, respectively (Fig. 3C and D); similarly, 23.26% and 51.43% of SMMA_5 and SMMA_5_TC cells remained viable after 45 min at 70°C in MST (Fig. 3E and F).

**Environmental solutes enhance thermal endurance, and even elicit nominal thermophilicity.** When SMMA_5 was incubated at 50°C for 12 h in MSTB, MSTL or MSTG, i.e., MST supplemented with 16 mM boron, 1 mM lithium, or 10 mM glycine-betaine, respectively, 14.4%, 29.2%, and 224% of the initial CFU counts were found to be present in the respective cultures (Fig. 6C and Fig. S2C). For SMMA_5_TC, 12 h of incubation at 50°C in MSTB, MSTL, and MSTG led to the CFU growths equivalent to 170%, 230%, and 370% of the initial counts, respectively (Fig. 6D and Fig. S2D). In the context of the MSTG data, it is noteworthy that after 12 h in unfortified MST at 50°C, the metabolically active but nondividing or hardly dividing cells of SMMA_5 and SMMA_5_TC were found to release 8 to 10 mM glycine-betaine to the spent medium. Corroboratively, the genomes of the two strains (analyzed below) were found to possess genes for betaine-aldehyde dehydrogenase and choline dehydrogenase, the key enzymes of the glycine-betaine biosynthesis pathway (27, 28).

After 45 min at 70°C in MSTB, MSTL, or MSTG, SMMA_5 retained 2.9%, 21.4%, and 5.5% of the initial CFU count, respectively (Fig. 6E and S2E). For SMMA_5_TC, 45 min incubation at 70°C in MSTB, MSTL, and MSTG led to the retention of 10.4%, 29.2%, and 15.8% of the initial CFU count, respectively (Fig. 6F and Fig. S2F). Furthermore, 37°C incubation of either strain for 12 h in MSTB, MSTL, and MSTG showed that boron, lithium, and glycine-betaine were not only nontoxic to them but also considerably stimulatory to their growth at this temperature (Fig. 6A and B; Fig. S2A and B).

The three environmental solutes also exhibited pronounced influence on the viability of SMMA_5 and SMMA_5_TC cells at 70°C (notably, at 50°C, almost 100% cell viability was already recorded via flow cytometry for both strains, without adding any of the three solutes to the MST medium). After incubation in MSTB, MSTL, and MSTG for 45 min at 70°C, 43.45%, 71.36%, and 55.17%, cells of SMMA_5 were found to remain viable, respectively (Fig. 7A to C). Likewise, after 45 min in MSTB, MSTL, and MSTG at 70°C, 66.04%, 89.31%, and 76.35% cells of SMMA_5_TC were found to remain viable, respectively (Fig. 7D to F).

**FIG 2** Legend (Continued)
of panel E is represented as a percentage of the corresponding 0 h CFU count. (G) CFU counts for SMMA_5_TC at 0 h and 12 h recorded at the different incubation temperatures; (H) each 12 h CFU count datum of panel G is represented as a percentage of the corresponding 0 h CFU count. All the data shown in this figure are averages obtained from three different experiments; error bars indicate the standard deviations of the data. Across the panels, and irrespective of the bacterium considered, all 0 h data are represented by blue columns, while the 12 h data obtained for the incubations at 37°C, 40°C, 45°C, and 50°C are represented by green, orange, yellow and pink columns, respectively. For all the incubation experiments carried out at 60°C and 70°C, 12 h CFU counts, and therefore the percentages of CFUs retained, were found to be zero; so there are no columns for these data.

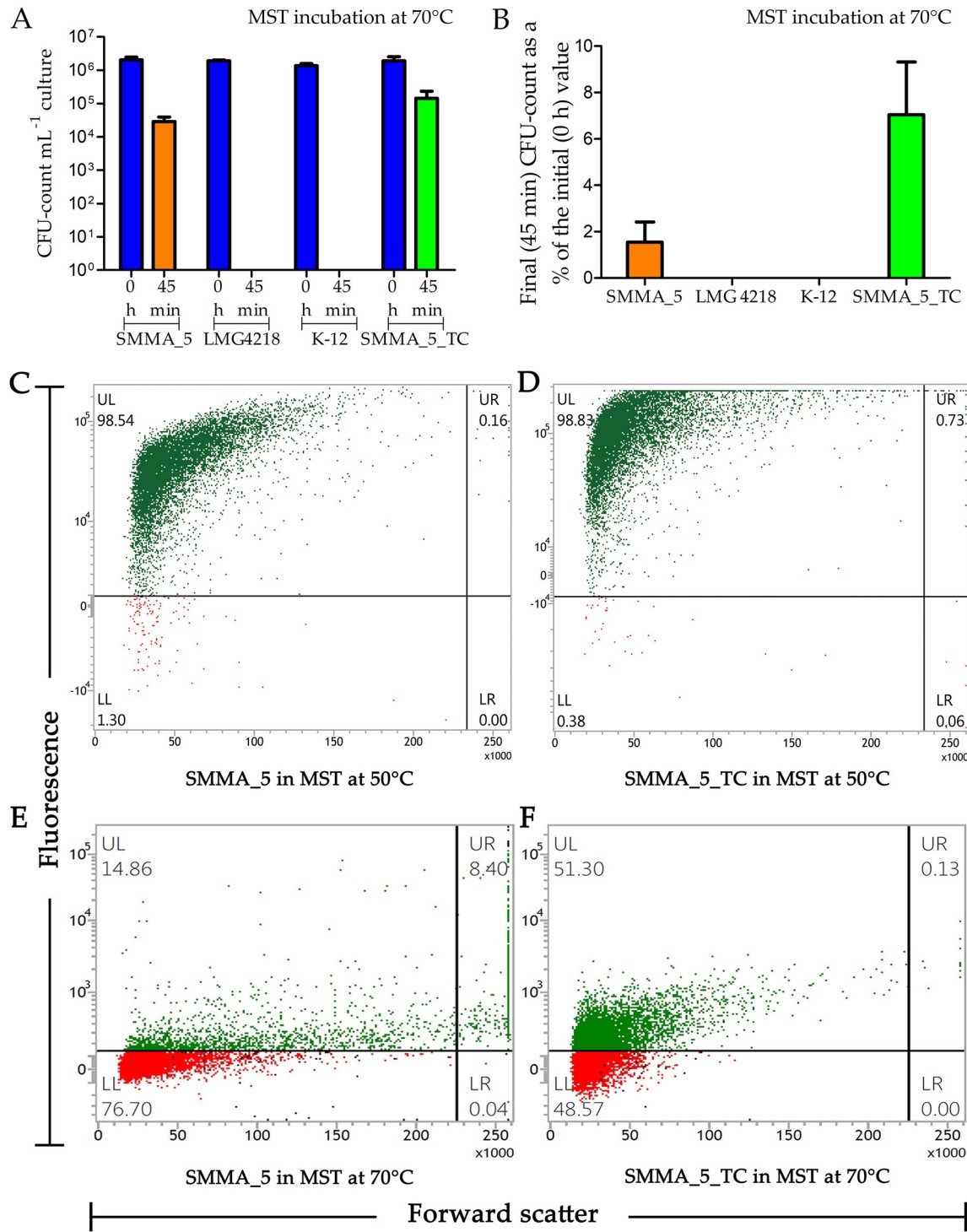

**FIG 3** CFU retention (A and B) by *Paracoccus* strains SMMA_5, LMG 4218, and SMMA_5_TC in MST, and *E. coli* strain K-12 in MSD, after 45 min of incubation at 70°C. Percentages of cells remaining viable (C to F) in MST cultures of SMMA_5 and SMMA_5_TC, after 12 h at 50°C and 45 min at 70°C, have also been shown. (A) Number of CFUs present mL$^{-1}$ of the different broth cultures, at 0 h and 45 min of incubation at 70°C; blue columns represent all the 0 h data across the cultures tested; orange and green columns represent the 45 min data for SMMA_5 and SMMA_5_TC; since no CFU was present in the LMG 4218 and K-12 cultures after 45 min, no column is shown for these data. (B) Percentages of the initial CFU counts that were retained in the SMMA_5, LMG 4218, SMMA_5_TC, and K-12 cultures after 45 min of incubation at 70°C (strain-wise color code for the columns is same as the one used in panel A). The data shown in A and B are averages obtained from three different experiments; error bars indicate the standard deviations of the data. (C to F) Flow cytometry-based dot plots showing the proportions of FDA-stained and FDA-unstained cells for (C) SMMA_5 after 12 h at 50°C, (D) SMMA_5_TC after 12 h at 50°C, (E) SMMA_5 after 45 min at 70°C, and (F) SMMA_5_TC after 45 min at 70°C. In panels C to F, fluorescence versus forward scatter data for 10,000 randomly taken cells have been plotted; green and red dots represent cells that were stained and not stained by FDA, respectively. Each flow cytometry-based experiment was repeated for two more occasions and in every instance <2% deviations were observed from the values shown here for the proportions of FDA-stained and FDA-unstained cells.

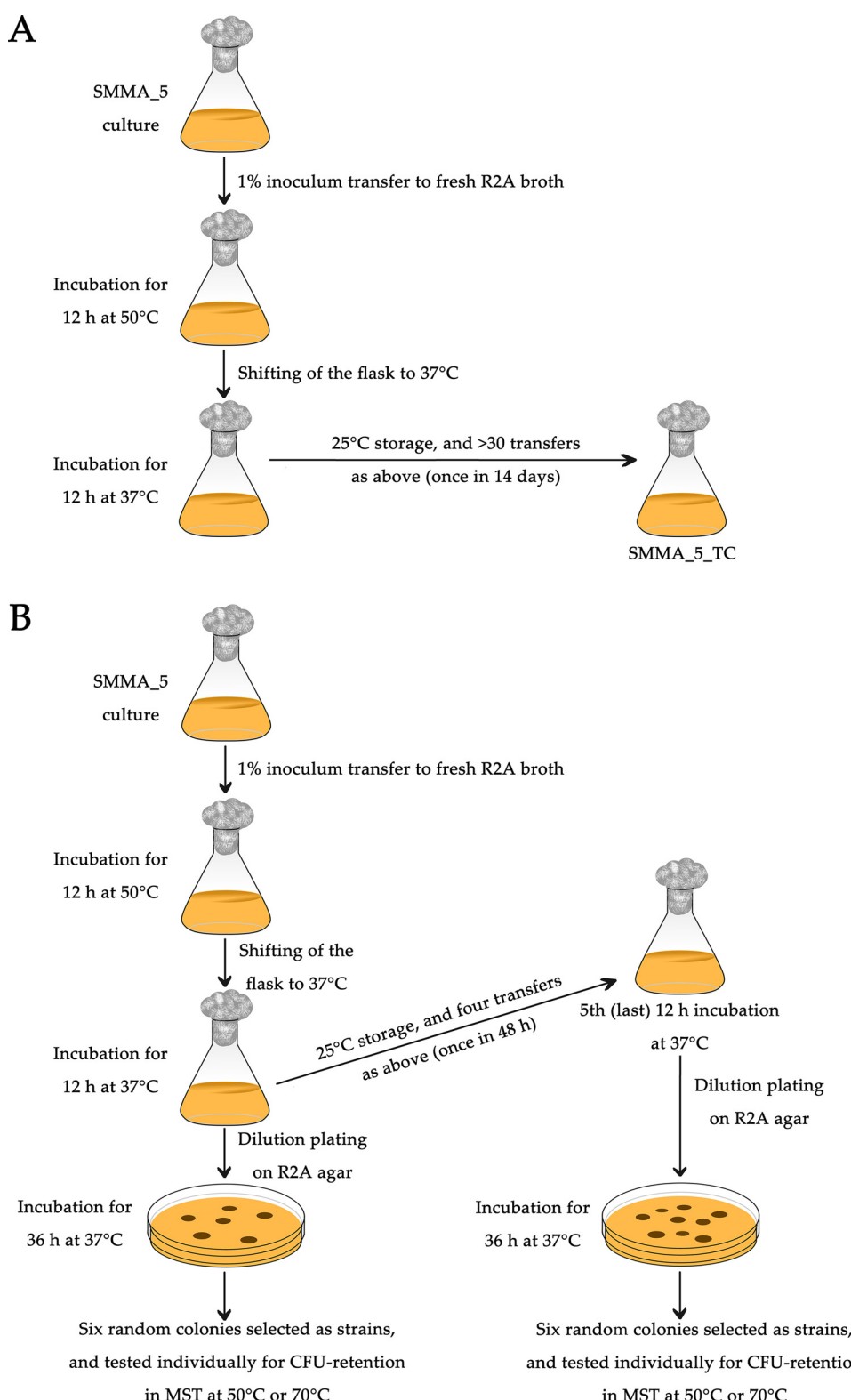

**FIG 4** Schematic diagrams showing the procedure followed for thermal conditioning of SMMA_5. (A) Steps followed in the creation of SMMA_5_TC. (B) Steps followed in raising sibling strains of SMMA_5 after one or five cycles of thermal conditioning.

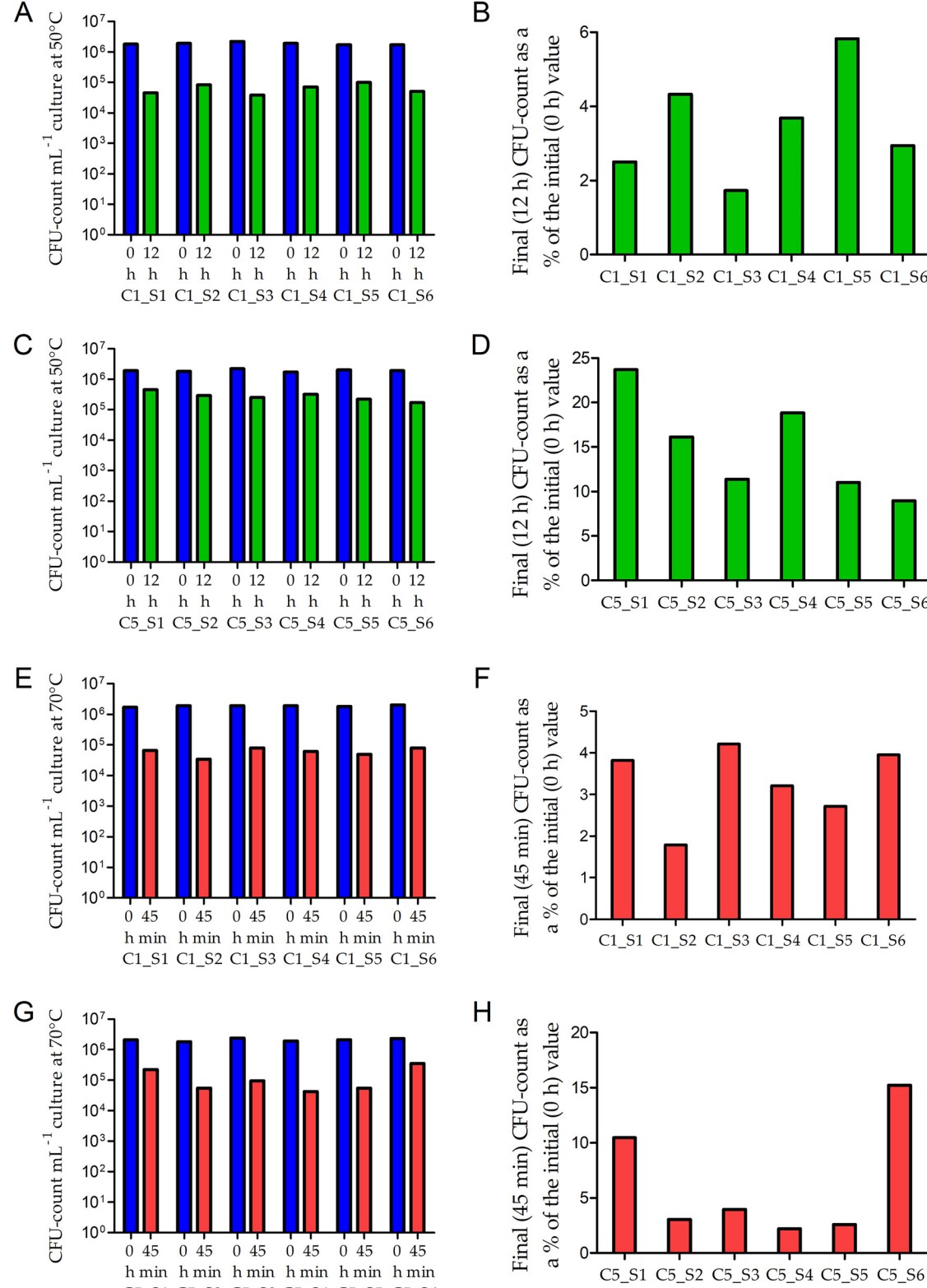

**FIG 5** CFU retention at 50°C (A to D) and 70°C (E to H) by multiple sibling strains of SMMA_5 generated after one (A, B, E and F) or five (C, D, G and H) cycles (generations) of thermal conditioning. (A) CFU counts at 0 h and 12 h in MST at 50°C for the six sibling strains of SMMA_5 (C1_S1 to C1_S6) that were raised via a single cycle of thermal conditioning. (B) Final (12 h) CFU count of each sibling strain shown in panel A, as a percentage of the corresponding initial (0 h) CFU count. (C) CFU counts at 0 h and 12 h in MST at 50°C for the six sibling strains of SMMA_5 (C5_S1 to C5_S6) that were raised via five cycles of thermal conditioning. (D) Final (12 h) CFU count of each sibling strain shown in panel C, as a percentage of the corresponding initial (0 h) CFU count. (E)

**Oligotrophy helps endure high temperature.** When incubated in R2A for 12 h at 37°C, both SMMA_5 and SMMA_5_TC exhibited >10$^4$ times increase in CFU count (Fig. 8A and B; Fig. S3A and B); comparable growth was recorded for the two strains in Luria broth after 12 h at 37°C (data not shown). After 12 h at 50°C in R2A, SMMA_5 and SMMA_5_TC retained 2% and 2.5% of the initial CFU counts, respectively (Fig. 8C and D; Fig. S3C and D), but after 12 h at 50°C in Luria broth, neither strain retained any CFU. Likewise, after 45 min at 70°C in R2A, SMMA_5 and SMMA_5_TC retained 0.5% and 1.0% of the initial CFU counts, respectively (Fig. 8E and F; Fig. S3E and F), but their Luria broth cultures retained no CFU after 45 min at 70°C.

At 37°C, 12 h of incubation in 0.1×, 0.01×, and 0.001× R2A resulted in progressively lower cellular growth yields, compared to what was recorded in R2A (1×), for both SMMA_5 and SMMA_5_TC (Fig. 8A and B; Fig. S3A and B).

After 12 h of incubation at 50°C in 0.1×, 0.01×, and 0.001× R2A, SMMA_5 retained 2.7%, 15.6%, and 33.3% of the initial CFU count, respectively (Fig. 8C and Fig. S3C), whereas SMMA_5_TC retained 3.5%, 20.0%, and 53.8% of the initial CFU count, respectively (Fig. 8D and Fig. S3D).

After 45 min at 70°C in 0.1×, 0.01×, and 0.001× R2A, SMMA_5 cultures retained 2.6%, 9.4%, and 14.9% of the initial CFU count, respectively (Fig. 8E and Fig. S3E), whereas SMMA_5_TC retained 3.1%, 13.4%, and 28.5% of the initial CFU count, respectively (Fig. 8F and Fig. S3F).

**pH 7.5 is optimum for thermal endurance.** The optimum pH for CFU growth at 37°C was tested in R2A, while the same for CFU retention at 50°C and 70°C was tested in 0.001× R2A. The two chemoorganoheterotrophic media types were chosen based on their best support for CFU growth/retention at the respective temperatures (Fig. 8); pH optima were not determined in MST medium because its salt ingredients precipitate out at pH >7.0 and temperature ≥50°C. For both SMMA_5 and SMMA_5_TC, R2A having a pH of 7.5 supported maximum growth after 12 h at 37°C (Fig. S4A and B; Fig. S5A and B); likewise, 0.001× R2A with a pH of 7.5 supported maximum CFU retention after 12 h at 50°C (Fig. S4C and D; Fig. S5C and D), as well as 45 min at 70°C (Fig. S4E and F; Fig. S5E and F).

**Rapid growth is not sustainable amid high heat.** SMMA_5_TC, when grown at 70°C in 0.001× R2A having an initial pH 7.5 and fortified simultaneously with 4 mM Na$_2$B$_4$O$_7$.10H$_2$O, 1 mM LiOH.H$_2$O and 10 mM C$_5$H$_{11}$NO$_2$, exhibited a 1.4 times increase in CFU count (from 1.9 × 10$^6$ mL$^{-1}$ to 2.73 × 10$^6$ mL$^{-1}$, as averages of three different experiments where standard deviations of the data were <2% of the means) after 45 min of incubation, but after 1 h, no CFU (or for that matter FDA-stained cell) was left in the culture.

**Genomic uniqueness of the hot spring *Paracoccus*.** The 3.09 Mb and 3.06 Mb draft genomes of SMMA_5 and SMMA_5_TC, as represented in 283 and 247 contigs (all >200 bp long), had G+C contents of 65.78% and 65.83%, respectively (Table S1). SMMA_5 contained 3,234 potential open reading frames (ORFs) or coding sequences (CDSs), out of which 2,498 and 503 encoded known and unknown (hypothetical) proteins, respectively; 16S, 23S, and 5S rRNA genes were identified together with 49 tRNA genes. The completeness level of this genome was estimated to be 98.23% (Table S1), based on the presence of *Paracoccus*-specific conserved marker genes curated in the CheckM database. The draft genome of SMMA_5_TC, on the other hand, was found to include 3,163 potential CDSs, out of which 2,511 and 477 encoded known and unknown proteins, respectively; 16S, 23S, and 5S rRNA genes were identified alongside 47 tRNA genes. Completeness of the SMMA_5_TC genome was 98.41% (Table S1),

**FIG 5** Legend (Continued)
CFU counts at 0 h and 45 min in MST at 70°C for the six sibling strains of SMMA_5 (C1_S1 to C1_S6) that were raised via a single cycle of thermal conditioning. (F) Final (45 min) CFU count of each sibling strain shown in panel E, as a percentage of the corresponding initial (0 h) CFU count; (G) CFU counts at 0 h and 45 min in MST at 70°C for the six sibling strains of SMMA_5 (C5_S1 to C5_S6) that were raised via five cycles of thermal conditioning. (H) Final (45 min) CFU count of each sibling strain shown in panel G, as a percentage of the corresponding initial (0 h) CFU count. Across the panels, 0 h, 12 h, and 45 min data are represented by blue, green, and red columns, respectively.

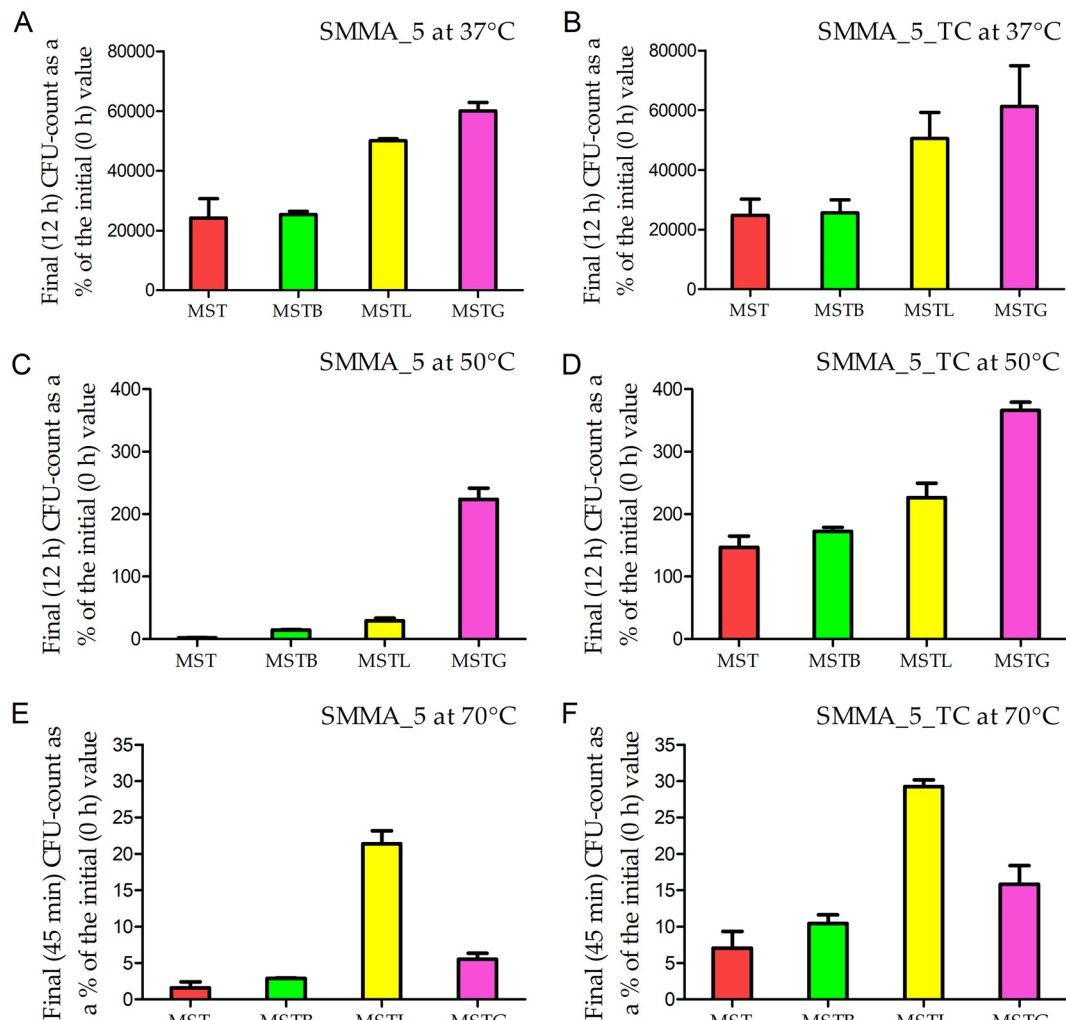

**FIG 6** Increase or decrease in the CFU count of SMMA_5 (A, C and E) and SMMA_5_TC (B, D and F) in MST, MSTB, MSTL, and MSTG media after incubation at 37°C (A and B), 50°C (C and D), and 70°C (E and F). (A) Final (12 h) CFU counts of SMMA_5 in the different media at 37°C, represented as percentages of the corresponding initial (0 h) CFU counts. (B) Final (12 h) CFU counts of SMMA_5_TC in the different media at 37°C, represented as percentages of the corresponding initial (0 h) CFU counts. (C) Final (12 h) CFU counts of SMMA_5 in the different media at 50°C, represented as percentages of the corresponding initial (0 h) CFU counts. (D) Final (12 h) CFU counts of SMMA_5_TC in the different media at 50°C, represented as percentages of the corresponding initial (0 h) CFU counts. (E) Final (45 min) CFU counts of SMMA_5 in the different media at 70°C, represented as percentages of the corresponding initial (0 h) CFU counts. (F) Final (45 min) CFU counts of SMMA_5_TC in the different media at 70°C, represented as percentages of the corresponding initial (0 h) CFU counts. All the data shown in this figure are averages obtained from three different experiments; error bars indicate the standard deviations of the data; the numerical values of all the CFU counts used to derive the above percentage values are shown in Fig. S2. Irrespective of the incubation temperature, data recorded for MST, MSTB, MSTL, and MSTG media are represented by red, green, yellow, and purple columns, respectively.

based on the presence of *Paracoccus*-specific marker genes. The average isoelectric point (pI) of all the putative proteins encoded by SMMA_5 and SMMA_5_TC were predicted as 6.86 and 6.87, respectively; corresponding values for all other *Paracoccus* species except *Paracoccus contaminans* were lower (Table S2; Fig. S6).

Pan genome analysis involving SMMA_5, and 44 other *Paracoccus* species having nearly complete genome sequences available in the GenBank database (Table S1), revealed 822 core genes for the genus. While SMMA_5 had 1,742 accessory, 335 unique, and 30 exclusively absent genes, the corresponding numbers in the comparators ranged between 1,528 and 3,451, 159 and 817, and 0 and 45 (Table S3). Of the 335 unique genes of SMMA_5 (Table S4), 186 encoded hypothetical proteins, whereas 149 coded for putative functional proteins. Of the latter 149, 83 belonged to clusters of

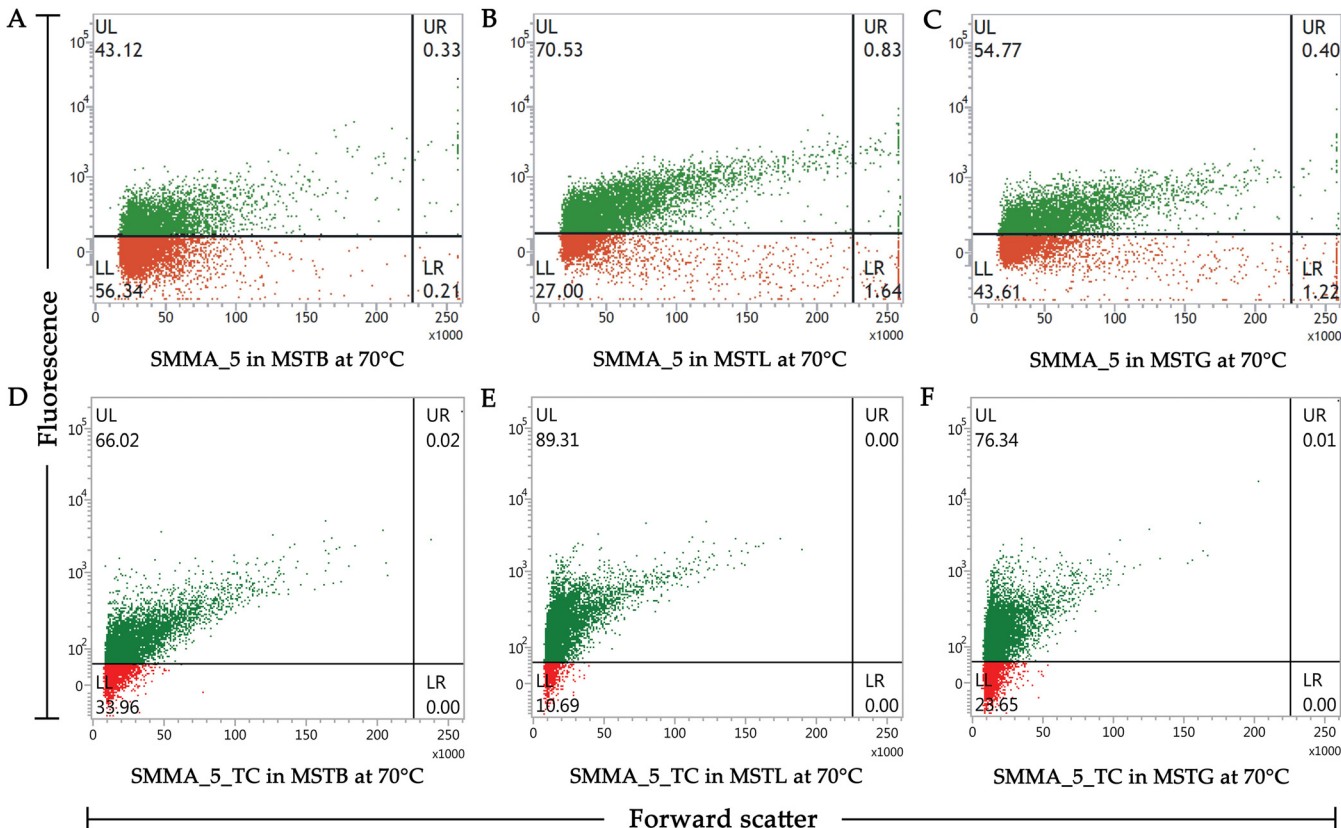

**FIG 7** Flow cytometry-based dot plots showing what percentage of cells remained viable (i.e., the proportion of FDA-stained and FDA-unstained cells) for SMMA_5 (A to C) and SMMA_5_TC (D to F), after 45 min of incubation at 70°C, in MSTB (A and D), MSTL (B and E), and MSTG (C and F). In each panel, fluorescence versus forward scatter data for 10,000 randomly taken cells have been plotted; green and red dots represent cells that were stained and not stained by FDA, respectively. Each experiment was repeated for two more occasions and in every instance <2% deviations were observed from the values shown here for the proportions of FDA-stained and FDA-unstained cells.

orthologous genes (COGs), which in turn were ascribable mostly to the COG categories DNA replication, recombination, and repair (13 genes); transcription (10 genes); secondary metabolites biosynthesis, transport, and catabolism (10 genes); inorganic ion transport and metabolism (9 genes); and cell wall/membrane/envelope biogenesis (8 genes).

The metabolic potentials of SMMA_5 were compared with those of its phylogenomically closest *Paracoccus* species (those 14 entities which occupied the same clade as SMMA_5 in the phylogenomic tree shown in Fig. 1) on the basis of their gene content under different COG categories. Two-way clustering of the genomes analyzed and COG categories identified revealed close functional relationships of SMMA_5 with *P. dinitrificans* and *P. thiocyanatus* on one hand, and *P. aminophilus* and *Paracoccus sulfurioxidans* on the other (Fig. 9A). Notably, in the phylogenomic tree, the first two species were closely related to SMMA_5, but the second pair diverged widely (Fig. 1). *P. pantotrophus*, *P. yeei*, *P. kondratievae*, *P. aminovorans*, and *P. lutimaris* showed the next closest relationships with SMMA_5 in the two-way dendrogram (Fig. 9A), but out of these five species only the first four shared the same branch with SMMA_5 in the phylogenomic tree whereas *P. lutimaris* was in a completely distinct branch (Fig. 1). *P. versutus*, which was phylogenomically closely related to SMMA_5 (Fig. 1), diverged widely from the new isolate in terms of gene-content under different COG categories (Fig. 9A). Statistical analysis further revealed that the two COG categories, amino acid transport and metabolism, and cell wall/membrane/envelope biogenesis, were enriched in SMMA_5 only, whereas lipid transport and metabolism was enriched in SMMA_5 as well as *P. aminovorans* (Fig. 9B).

Overall enrichment of the three COG categories notwithstanding, the numbers of unique SMMA_5 genes ascribable to these metabolisms were not too many. When

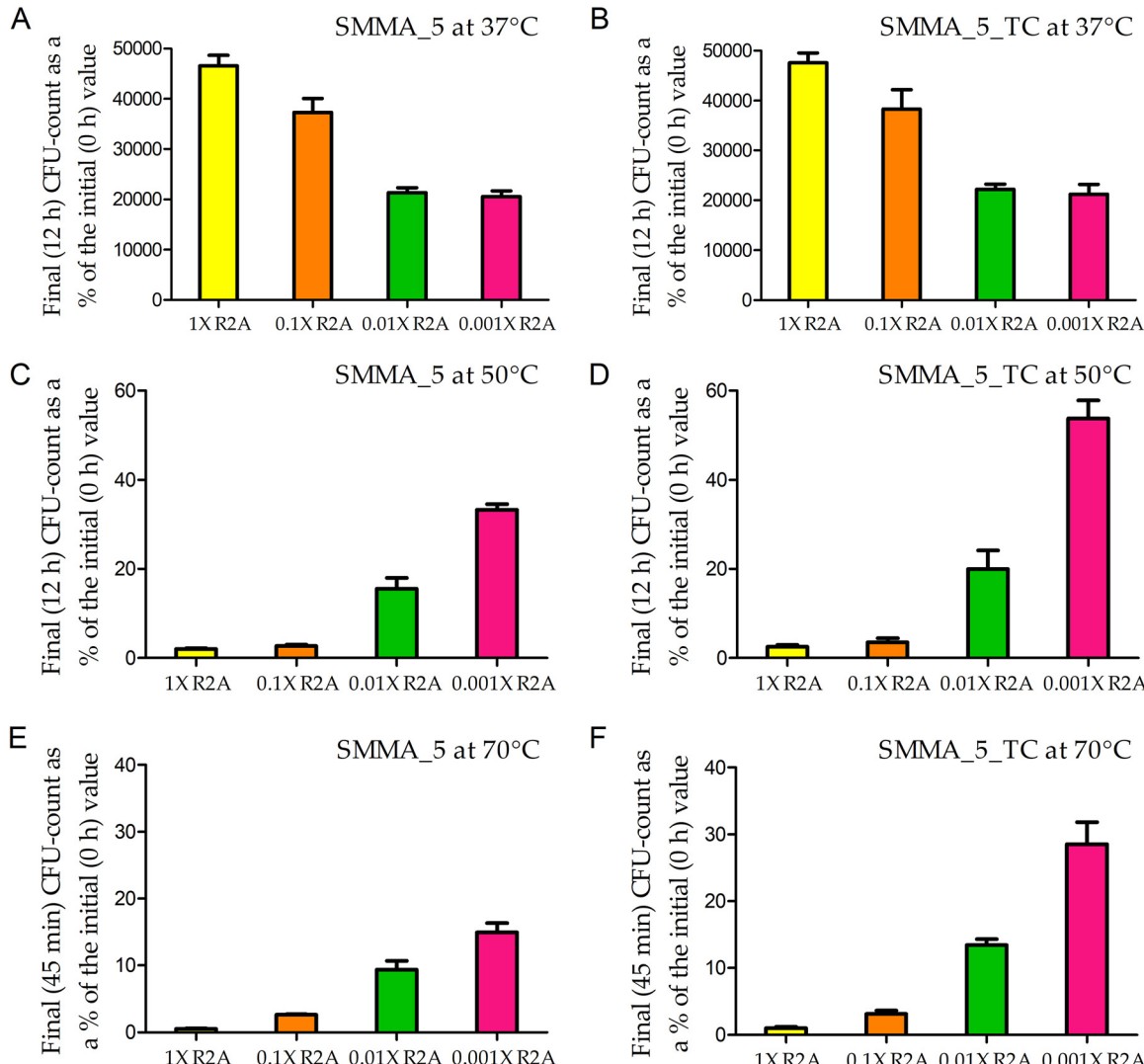

**FIG 8** Increase or decrease in the CFU count of SMMA_5 (A, C and E) and SMMA_5_TC (B, D and F) in 1× R2A, 0.1× R2A, 0.01× R2A, and 0.001× R2A media after incubation at 37°C (A and B), 50°C (C and D), and 70°C (E and F). (A) Final (12 h) CFU counts of SMMA_5 in the different media at 37°C, represented as percentages of the corresponding initial (0 h) CFU counts. (B) Final (12 h) CFU counts of SMMA_5_TC in the different media at 37°C, represented as percentages of the corresponding initial (0 h) CFU counts. (C) Final (12 h) CFU counts of SMMA_5 in the different media at 50°C, represented as percentages of the corresponding initial (0 h) CFU counts. (D) Final (12 h) CFU counts of SMMA_5_TC in the different media at 50°C, represented as percentages of the corresponding initial (0 h) CFU counts. (E) Final (45 min) CFU counts of SMMA_5 in the different media at 70°C, represented as percentages of the corresponding initial (0 h) CFU counts. (F) Final (45 min) CFU counts of SMMA_5_TC in the different media at 70°C, represented as percentages of the corresponding initial (0 h) CFU counts. All the data shown in this figure are averages obtained from three different experiments; error bars indicate the standard deviations of the data; the numerical values of all the CFU counts used to derive the above percentage values are shown in Fig. S3. Irrespective of the incubation temperature, data recorded for 1× R2A, 0.1× R2A, 0.01× R2A, and 0.001× R2A media are represented by yellow, orange, green, and pink columns, respectively.

considered in relation to the 14 phylogenomically closest relatives, SMMA_5 had 13, 9, and 8 unique genes ascribed to amino acid transport and metabolism, lipid transport and metabolism, and cell wall/membrane/envelope biogenesis, respectively (Table S5); in contrast, compared with all 44 *Paracoccus* species having nearly complete genome sequences available, these numbers became 2, 5, and 8, respectively (Table S4). Furthermore, no distinctive skew in favor of the catabolic or the anabolic halves of amino acid or lipid metabolism was identified for the SMMA_5 genome; there was also no special pathway for the biosynthesis or transformation of amino acids or lipids. The relative abundances of the 20 individual amino acids (expressed as the percentage of all amino acids present) were found to remain conserved across the putative (*in silico*

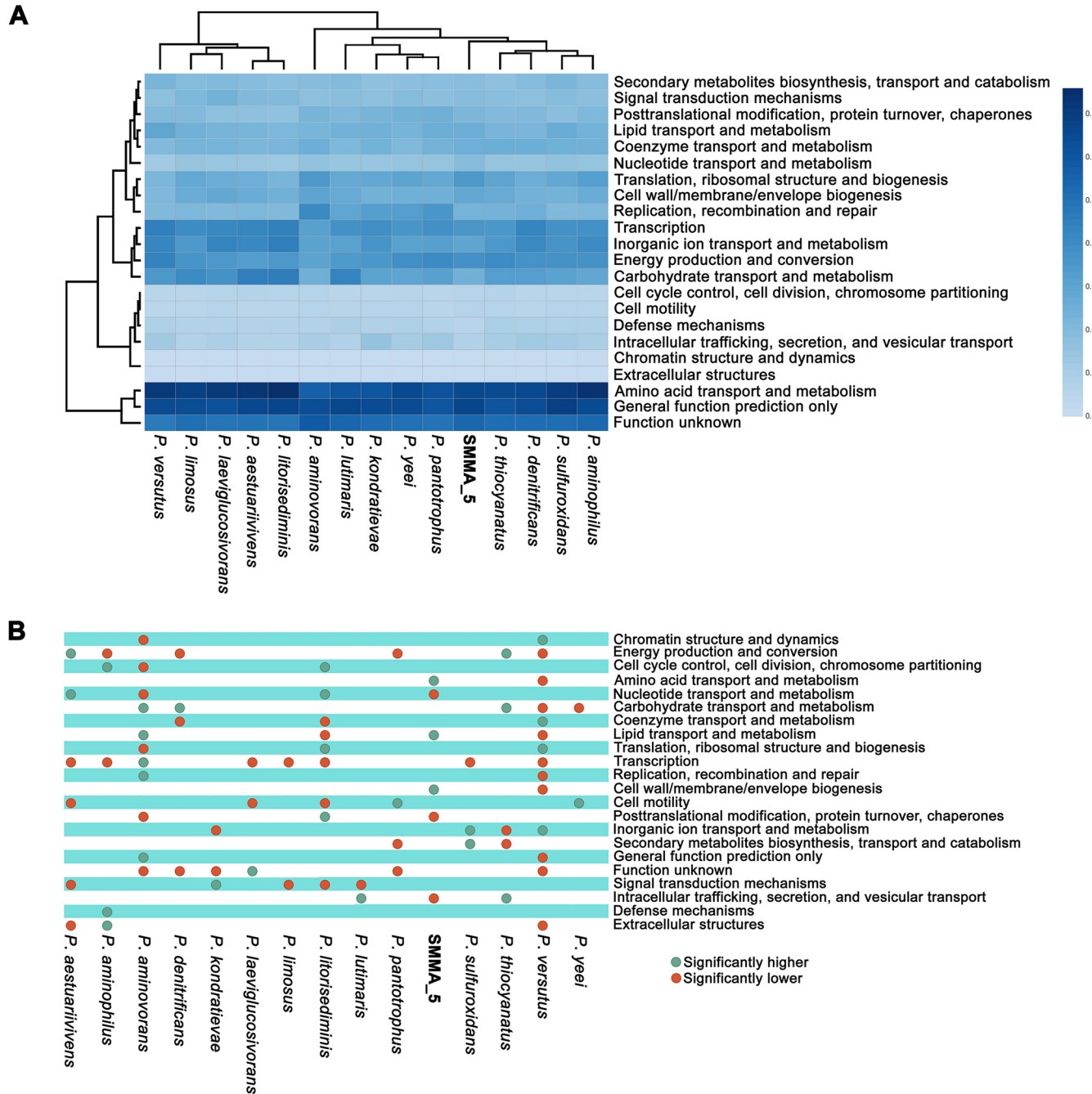

**FIG 9** Functional analysis of the genomes of SMMA_5 and its 14 phylogenomically closest *Paracoccus* species. (A) Heat map comparing the richness of the functional/metabolic categories across the genomes, determined in terms of the number COGs that are ascribed to the categories in individual genomes. A two-dimensional clustering is also shown, involving the 15 bacterial genomes on one hand and the 22 functional categories of COGs on the other. Color gradient of the heat map varies from high (deep blue) to low (faint blue), through moderate (light blue) richness of the categories across the genomes. (B) Statistically significant, high (green circles) or low (red circles) richness of the functional categories of COGs, detected across the genomes via Chi-Square test with $P < 0.001$.

translated) proteomes of SMMA_5 and its 14 closest phylogenomic relatives, with the standard deviations for the relative abundances of all the amino acids except alanine being <2% of their corresponding means (Table S5). Nevertheless, SMMA_5, compared with its 14 closest phylogenomic relatives, had the highest (or just-near highest) relative abundances of cysteine, histidine, glutamine, and arginine in the putative whole proteome. In contrast, relative abundances of glutamic acid, phenylalanine and lysine were lowest in SMMA_5. Only 3 out of the 14 closest phylogenomic relatives had a

higher relative abundance of proline in their putative proteomes, compared to SMMA_5; and 10 out of the 14 closest phylogenomic relatives had a higher relative abundance of glycine, compared to SMMA_5.

**Genomic aspects potentially linked to thermal adaptation.** The genomes of both SMMA_5 and SMMA_5_TC encompassed a large repertoire of heat shock proteins and molecular chaperones. These included (i) one copy each of the cochaperone GroES, the heat shock protein HspQ, the heat-inducible transcriptional repressor HrcA, the Hsp33 family molecular chaperone HslO, the cochaperone GrpE, and the RNA polymerase sigma 32 factor RpoH; (ii) two copies each of the genes encoding the chaperonin GroEL, Hsp20 family protein, and the Hsp70 family protein DnaK; and (iii) three copies of the molecular chaperone DnaJ (Hsp40). Genes for heat shock protein Hsp90 and RNA polymerase sigma E factor, which are typically associated with heat stress management in *E. coli*, were neither present in SMMA_5 and SMMA_5_TC, nor in any of the 14 species that were phylogenomically closest to the hot spring *Paracoccus* (Table S6). On the other hand, the presence of genes for the redox-regulated, Hsp33-family chaperone HslO distinguished the genomes of SMMA_5, SMMA_5_TC, and most of the comparator *Paracoccus* species, from those of *E. coli*, *P. denitrificans*, *P. laeviglucosivorans*, *P. lutimaris*, *P. sulfuroxidans*, *P. thiocyanatus*, and *P. versutus* (HslO is present in most thermophilic bacteria to protect thermally or oxidatively damaged proteins from irreversible aggregation). Apart from the heat shock proteins and chaperones, the genomes of SMMA_5 and SMMA_5_TC encoded several other proteins that can be potentially decisive in thermal adaptation by means of their regulated expression (Table S7).

The draft genomes of SMMA_5 and SMMA_5_TC showed 99.9748% sequence similarity over a total alignment length of 3,061,077 nucleotides (unaligned portions of the two genomes were attributable to their incomplete sequencing, rather than the presence of unique loci). Of the total 43 instances of single nucleotide polymorphism (SNP) detected across the alignment, 18 were in protein-coding genes, 18 were in frameshifted pseudogenes, whereas seven were in noncoding regions (Table S8). Out of the 18 nucleotide substitutions detected in protein-coding genes, 13 were nonsynonymous (ns) SNPs while five were synonymous (silent) mutations (sSNPs). Of the 13 nsSNPs, again, nine and four involved transitions and transversions, respectively, while seven involved such radical replacements of amino acids (e.g., polar to nonpolar, or aromatic-group-containing to alkyl-group-containing, amino acids, or vice versa) that may lead to changes in the corresponding protein structure and function (Table S8).

## DISCUSSION

A complex set of geobiological and biophysical factors worked in conjunction with specialized genomic attributes to confer thermal endurance, or even nominal thermophilicity, to the hot-spring-dwelling relative of mesophilic *Paracoccus*. While the thermally unconditioned isolate SMMA_5 had an intrinsically higher frequency of CFU retention at 50°C and 70°C, compared with other *Paracoccus* (and *E. coli*), its sibling SMMA_5_TC, which had been thermally conditioned for more than 30 transfer cycles, showed either growth or yet higher frequencies of CFU retention at these temperatures. Corroboratively, thermal conditioning for only five transfer cycles brought about, on an average, eight and four times increases in the CFU retention frequencies of SMMA_5 at 50°C and 70°C, respectively. Geochemical and microbial solutes further improved the growth/survival efficacies at 50°C and 70°C, as did extreme oligotrophy at pH 7.5. Overall, it seems that during their residence in the solute-poor, circum-neutral pH, and near-boiling water of Lotus Pond (7, 12), cell populations of this novel *Paracoccus* had already acquired such heritable structural and functional attributes which are conducive for the mitigation of high heat. *Ex situ*, the potential cellular inheritances conferring heat endurance (or nominal thermophilicity) were on the wane in the thermally unconditioned culture (SMMA_5), while the same were retained relatively better in the thermally conditioned variant (SMMA_5_TC).

**Prior experience of high temperature enhances heat endurance or confers nominal thermophilicity.** Preexposure to sublethal stress conditions is known to enhance a microorganism's resilience to higher levels of stress (32). Bacteria are endowed with diverse memory mechanisms, including those which involve the inheritance and propagation of epigenetic states associated with adaptive advantages (33, 34). Of the various metabolic systems that are known to involve heritable epigenetic memory, ion-channel-mediated signaling and ion flux machineries (35–37) appear to be pertinent to thermal endurance in view of the observed influence of environmental solutes on growth and survival at high temperatures. Notably, however, the genome of SMMA_5, compared with those of its closest phylogenomic relatives, contained fewer genes for channel proteins and porins/aquaporins; ABC transport systems; cobalt, copper, magnesium and/or nickel transporters; uni-/sym-/antiporters; ECF class transporters; Ton and Tol transporters; and TRAP transporters (Table S9). Concurrently, out of the 335 genes unique to SMMA_5, compared with all other *Paracoccus* species for which whole-genomes have been sequenced, only nine were ascribable to inorganic ion transport/metabolism (Table S4). Thus, any potential involvement of membrane transport in the thermal adaptation of SMMA_5 ought to be at the expression level, rather than being dictated directly by the genome content.

Under almost all the culture conditions tested, the standard deviations of the CFU retention data for SMMA_5_TC were higher than those for SMMA_5. Concurrent to this trend, sibling SMMA_5 strains generated via few cycles of thermal conditioning did not show equal frequencies of CFU retention at any of the high temperatures tested. These data collectively indicated that in the hot-spring-dwelling *Paracoccus*, thermal conditioning via potential epigenetic trait retention could be a multifactorial and asymmetrically inherited (34) process that involves a "bet and hedging" strategy, where heterogeneous populations of phenotypically distinct but genotypically identical cells are formed and maintained by a bacterium to ensure that one subpopulation or the other can always withstand and adapt to unknown environmental challenges of the future (38).

**Role of environmental solutes in enhancing thermal endurance or eliciting nominal thermophilicity.** Out of the three environmental solutes tested, the alkali metal lithium was most effective in increasing the frequency of CFU retention, as well as the percentage of viable cells, at 70°C, for both SMMA_5 and SMMA_5_TC, compared to their respective phenotypes in unfortified MST. At 50°C too, lithium fortification of MST increased the frequency of CFU retention and CFU growth for SMMA_5 and SMMA_5_TC, respectively. These phenotypes were consistent with previous observations where *Listeria monocytogenes* exposed to 62.8°C for 10 to 20 min was recovered after 48 to 144 h, by adding 7 g $L^{-1}$ LiCl to the 30°C revival cultures (18). The thermoprotective effect of lithium could be linked, but not necessarily restricted, to its activities at the periphery of cells. $Li^+$ efficiently forms hydrogen bonds with nearby water molecules, as well as charged surfaces such as cell membranes, thereby modifying their intrinsic van der Waals, and electrostatic, forces (39). $Li^+$ also competes for ligand-binding sites of biomacromolecules that are otherwise reserved for $Na^+$ or $Mg^{2+}$ (40). By virtue of these biophysical maneuvers, $Li^+$ cannot only stabilize biomacromolecules but also enhance the permeability of those constituting cellular membranes, which in turn can enhance the entry of other thermo-/osmo-protective solutes (41, 42).

The compatible solute glycine-betaine also significantly increased the frequency of CFU retention, as well as the percentage of viable cells, at 70°C, for both SMMA_5 and SMMA_5_TC. Furthermore, at 50°C, glycine-betaine fortification of MST brought about an unprecedented CFU growth for SMMA_5, and a small but definite increase in the existing CFU growth of SMMA_5_TC. Glycine-betaine is taken-up/synthesized in large amounts by halophilic/halotolerant microbes as their secondary response to high external salt concentrations (28). However, besides acting as osmotic balancers, compatible solutes can also impart a general stabilizing effect on biomacromolecules (43, 44). As the highly soluble glycine-betaine molecules accumulate within the cell, their preferential

exclusion from the immediate hydration sphere of proteins leads to a nonhomogeneous distribution within the cell-water, thereby causing a thermodynamic disequilibrium (45). This disequilibrium is minimized via reduction in the volume of cell-water from which the solute is excluded; this, in turn, is achieved via reduction in the surface area or volume of the proteins through increases in subunit assembly and stabilization of secondarily folded tertiary structures (45). Such biophysical maneuvers at the molecular level can add up at the cellular level to confer tolerance against any physicochemical stressor that tends to disrupt the system by increasing the disorder within biomacromolecules (45, 46).

Supplementing MST with boron also led to a small but definite increase in the frequency of CFU retention, as well as the percentage of viable cells, at 70°C, for both SMMA_5 and SMMA_5_TC. At 50°C, boron addition to MST marginally increased the frequency of CFU retention and CFU growth for SMMA_5 and SMMA_5_TC, respectively. Boron possesses unique bonding properties that can expand the biophysical functions of macromolecules, including their modes of Lewis acidity. *In vitro* insertion of boron to proteins in the form of boronoalanine has been shown to allow dative bond-mediated and site-dependent protein Lewis acid-base-pairing, which in turn can generate new functions, including stability in the face of high temperature or proteolytic attack (47). Furthermore, it is theoretically not impossible that boron bonds with membrane fatty acids to form organoborane complexes which in turn provide cross-linkages, and thereby stability, for structures such as cellular membranes.

**Oligotrophy as a strategy for survival at high temperature.** After 12 h of incubation at 37°C, cellular growth yield of both SMMA_5 and SMMA_5_TC was higher in $1\times$ R2A than in any of the dilution grades of the medium. But, after 12 h of incubation at 50°C, as well as after 45 min at 70°C, all the dilution grades of R2A supported the retention of higher CFU percentages than undiluted R2A, for both SMMA_5 and SMMA_5_TC. On the other hand, compared with the corresponding MST phenotype, CFU retention efficacy in most of the dilution grades of R2A was higher for SMMA_5, but not SMMA_5_TC, when incubation was carried out for 12 h at 50°C, but higher for both SMMA_5 and SMMA_5_TC, when incubation was carried out for 45 minutes at 70°C.

Thus, SMMA_5 and SMMA_5_TC were found to be best adapted to cope 50°C temperature under oligotrophic and chemolithoautotrophic conditions, respectively, whereas both strains were best adapted to cope 70°C under oligotrophic condition. These findings, in concurrence with the differential effects exhibited by the environmental solutes on the endurance of different temperature levels, indicated that the hot spring *Paracoccus* potentially modulates its metabolic response to thermal stress with increasing temperature, and such response regulation plausibly involves a differential control of the global transcriptome, orchestrated over prolonged thermal conditioning. Overall, the survival strategy amid high heat appears to be aimed at slowing down growth, and prioritizing cell-system maintenance over proliferation.

Oligotrophy also affords collateral advantages in combating a wide variety of stresses, including high heat (48). It entails a constant ability to take up whatever little nutrients are available in the cells' chemical milieu by using low-specificity, high-affinity, and low-activation-energy transport systems (49). However, in the context of a hot spring habitat, constant nutrient uptake via low-specificity transport systems implies that hydrogen peroxide ($H_2O_2$) and other reactive oxygen species (ROS) abundant in hydrothermal environment also get unrestricted entry into the cell, as do the potentially thermoprotective inorganic solutes and small organic molecules. In the process, cells develop intrinsic resistance against $H_2O_2$/ROS toxicity via enhanced expression of genes responsible for membrane detoxification, and protein repair and maintenance (50), which in turn protect the cells' macromolecules against the destabilizing effects of heat.

**Genomic underpinnings of thermal endurance.** The hot spring *Paracoccus* possessed many genes for the quality management of DNA and protein under stress conditions (Table S6 and S7). At the same time, the genome of SMMA_5, compared with those of the phylogenomically closest *Paracoccus*, was found to be relatively enriched in genes for cell wall/membrane/envelope biogenesis, and transport and metabolism

of amino acids and lipids. Notably, however, most of the SMMA_5 genes (and putative biochemical pathways) identified under these metabolic categories were also there in most of the *Paracoccus* species compared. That said, the overall combination of genes present in the SMMA_5 repertoire could still be instrumental in thermal adaptation via intricate regulation at the expression level.

Thermal conditioning, or the lack of it, appears to have a small but definite effect on the genome. This is consistent with the fact that temperature influences the rate and nature of mutations, and thereby natural selection, evolution, and biogeography of a microorganism (51). Of the total 43 SNPs identified across the aligned portions of the SMMA_5_TC and SMMA_5 genomes, 18 were in protein-coding genes. Out of these 18, again, 13 and five were nsSNPs and sSNPs, respectively, which is a ratio that is potentially indicative of adaptive evolution (52) or relaxed selective constraints (53) in the thermally conditioned (SMMA_5_TC) lineage. Of the 13 nsSNPs detected in protein-coding genes, nine and four involved transition and transversion mutations, respectively, while seven resulted in radical amino acid replacements having potentials to change protein structures and functions in such ways that can impact activities/phenotypes (Table S8). While radical amino acid replacements often turn out to be beneficial mutations that enhance protein functions (54), those encountered in the putative selenide, water dikinase protein SelD, sulfate adenylyltransferase subunit 2 CysD, and DNA polymerase III subunit gamma/tau, appear to hold potential implications for the enhanced thermal endurance or nominal thermophilicity of SMMA_5_TC.

Bacteria have a large repertoire of selenocysteine-containing proteins, and both SelD and CysD play vital roles in their biosynthesis. While the role of selenoproteins in cellular stress management is well documented (55), CysD is also involved in the first step of sulfite formation during the assimilatory reduction of sulfate to sulfide (56). Since ROS is a major driver of cell death during thermal stress (57, 58), the highly reduced sulfide species produced during sulfate reduction can potentially defend microbial cells amid high heat (59). The efficiency of these processes may well be enhanced by potentially modified homologs of SelD and CysD.

Gamma and tau subunits of DNA polymerase III, the stoichiometric components of the replicative complex, are engaged in loading the processivity clamp beta (60, 61). As hyperthermia inhibits DNA replication by either slowing down or arresting the replication forks depending on the temperature level and cell type (62), it is not unlikely that the DNA polymerase III subunit gamma/tau of SMMA_5_TC has potential structural advantages for high-temperature functioning.

**The putative proteome of the hot spring *Paracoccus* could be biophysically adapted to the environment.** Microbial species adapted to distinct temperature regimes are known to have different amino acid frequencies in their proteomes, even as the pattern of amino acid distribution varies significantly from the cores to the surfaces of proteins (63). In our analyses involving SMMA_5 and its 14 phylogenomically closest *Paracoccus* species, relative abundances of all 20 amino acids were found to be essentially conserved across the putative proteomes considered (Table S10). Furthermore, the cumulative frequency of isoleucine, valine, tyrosine, tryptophan, arginine, glutamic acid, and leucine in the proteome, which is said to correlate positively with the optimum growth temperature of microorganisms (64), was lower in SMMA_5, compared to 7 out of the 14 closest relatives (Table S10). All these aspects of the SMMA_5 proteome, in the context of the comparator entities, suggested that the habitat-compelled thermal adaptation of the hot spring *Paracoccus* has not been so longstanding as to have the genome/proteome radically evolved in sync with the environmental stressor. That said, incipient genome-based signatures of thermal adaptation could be construed from the high relative abundances of cysteine and proline, and the relatively lower frequency of glycine, in the putative proteome of SMMA_5, compared with the closest phylogenomic relatives.

Disulfide bridges between cysteine residues minimize the freedom of movement of peptide chains, and in doing so stabilize extracytoplasmic as well as cytoplasmic proteins (63, 65). Therefore, a small increase in the proportion of this most-infrequent amino acid in the whole proteome may well translate into enhanced thermal endurance.

Proline is relatively more abundant in proteins adapted to high temperatures. By virtue of being more rigid than other amino acids it can reduce the freedom of movement of peptide chains, minimize entropy, and thereby decrease the chances of high temperature induced protein unfolding; moreover, proline is hydrophobic, so can contact other hydrophobic amino acids in the core as well as the peripheral parts of the proteins; all these features add to the structural stability of the proteins (63).

As opposed to proline, the small amino acid glycine makes proteins more flexible, and decreases their structural rigidity, by virtue of lacking any side chain which can restrict the freedom of movement for the polypeptides that they are a part of (63). Thus, even a small decrease in the overall glycine richness of the proteome can potentially lead to a higher level of thermal endurance.

The pH of the chemical milieu is a key biophysical parameter for biomacromolecular stability, and thereby overall microbial adaptation to different temperature regimes. Consequently, it was imperative for the present study to investigate the pH optima for thermal endurance by the hot spring *Paracoccus*; we did so and found that a pH of 7.5 was best for growth at 37°C and CFU retention at 50°C and 70°C. With the elevation of temperature and/or increase in the time of heat exposure, conformational changes in the structure of proteins decrease their solubility and tend to precipitate them out of the solution. Solubility of most proteins increases up to the 40 to 50°C temperature range, and starts decreasing irreversibly beyond these levels, depending on the pH of the medium, and also its ionic strength (66). In this scenario, higher electrostatic repulsion between the molecules of a protein, compared with the degree of hydrophobic interactions between them, can promote solubility via enhanced interaction with the solvent. Furthermore, at pH values above their pI, net charge of proteins become negative, so repulsion between their molecules, and thereby their solubility (interactions with water), increases (66). In the context of thermal endurance by the hot spring *Paracoccus* it is noteworthy that (i) the average pI values predicted for all the putative proteins of SMMA_5 and SMMA_5_TC were 6.86 and 6.87, respectively, with 61.2% and 61.1% of all proteins of the two strains having pI values <6.86 and <6.87, respectively; (ii) in oligotrophic media, pH 7.5 was found to be optimum for growth at 37°C, and CFU retention at 50°C and 70°C, concurrent to which, 65.8% and 65.7% of all SMMA_5 and SMMA_5_TC proteins were predicted to have pI <7.5, respectively; and (iii) the measured pH of Lotus Pond's vent-water from where SMMA_5 was isolated ranged between 7.2 and 8.0, corresponding to *in situ* temperatures 85°C and 78°C (7).

In this way, for this microorganism, both *in vitro* pH optimum and the *in situ* pH of the habitat were above the average pI predicted for the total proteome. Moreover, these pH values, at high temperatures, represent further alkaline environments, which in the light of the above biophysical considerations appear to be all the more favorable for the conformational integrity, solubility, and therefore the functionality of the majority of proteins that are synthesized by the hot spring *Paracoccus*.

**Metabolic deceleration as central to thermal endurance.** A comparison between the generation times of SMMA_5 and *P. pantotrophus* LMG 4218, recorded at 37°C, 40°C, and 45°C (in MST medium), showed that with increasing temperature growth decelerated in both the strains, but the slowdown was less in SMMA_5, compared to LMG 4218. Albeit no growth was observed for either strain at 50°C under the routine culture conditions, a small increase in CFU count, reflective of extremely low growth rate, was observed for the hot spring *Paracoccus* alone after certain habitat-inspired manipulations were rendered in the strain's maintenance history and/or the chemical composition of the culture media. MST incubation of the hot spring *Paracoccus* at 50°C and 70°C, with or without lithium, boron, or glycine-betaine, showed that a greater proportion of cells always remain metabolically active and viable, compared to what percentage of cells remain ready to divide and form single colonies upon withdrawal of heat. Corroboratively, an event of rapid cell division at 70°C, in 0.001× R2A fortified simultaneously with lithium, boron, and glycine-betaine, proved to be deleterious to the survival of the entire population. All these phenotypes were reflective of the fact

that an overall deceleration of metabolism, underplaying cell division and aimed at achieving a cost-effective maintenance of the cell-system, was central to the thermal endurance strategy of the hot spring *Paracoccus*. Development of "viable but not readily culturable" states in high proportions under different stress conditions is a widespread phenomenon in bacteria (67, 68). At the level of gene expression, metabolic deceleration in the hot spring *Paracoccus* could involve intricate regulation of the genomic loci attributed to growth and cell division, oxidative stress management, quality control of biomacromolecules, maintenance of cell wall/envelope integrity, membrane permeability, transport, and energy production/utilization. Concomitantly, thermal conditioning, whether *in situ* or *in vitro*, could help the organism inherit those regulatory states of global gene expression which are central to metabolic slow-down and cell-system maintenance. Eventually, at the level of the biomacromolecules, mitigation of the disordering effects of heat requires such intrinsic entropy-minimizing mechanisms (1, 69) which can accomplish a fine balance between structural integrity (needed to avoid denaturation) and flexibility (necessary to preserve functionality). While the hot spring *Paracoccus* seems to possess innate biophysical contrivances for the stabilization of its macromolecular structures amid high heat (and thereby the lowering of the energy cost of cell-system maintenance), environmental solutes, on their part, appear to augment the cells' indigenous stability-conferring mechanisms. Future studies of transcriptomics, proteomics, and metabolomics, resolved along the dimensions of time, temperature, and biogeochemical variables are needed to fully conclude how hot spring mesophiles survive high temperatures over time and space.

## MATERIALS AND METHODS

**Bacterial strains, media, and culture conditions.** The facultative chemolithoautotroph SMMA_5 was isolated, as described previously, via enrichment in MST medium that essentially contained a modified basal and mineral salts (MS) solution supplemented with thiosulfate (7, 12). The strain was deposited in the Microbial Type Culture Collection and Gene Bank (MTCC), Chandigarh, India with the public accession number MTCC 12601. Since its isolation, SMMA_5 was maintained in R2A medium by growing the culture at 37°C and storing at 4°C, with a standard transfer interval of 28 days. From the main culture, the thermally conditioned variant SMMA_5_TC was created and maintained in the following way. A freshly inoculated R2A broth culture of SMMA_5 was first incubated at 50°C for 12 h and then shifted to 37°C for another 12 h. While no turbidity appeared in the spent medium over the 50°C incubation phase, growth corresponding to an $OD_{600}$ of ~0.6 was obtained at the end of 12 h at 37°C. The culture flask was then stored at 25°C and transferred following the above procedure after an interval of 14 days (Fig. 4A). The comparator strains, *P. pantotrophus* LMG 4218 and *E. coli* K-12, were maintained in Luria broth by growing the cultures at 37°C and storing at 4°C, with a standard transfer interval of 28 days.

**Phylogenetic identification based on 16S rRNA or 92 conserved marker gene sequences.** The 16S rRNA gene of the new isolate SMMA_5 was PCR-amplified and sequenced using the *Bacteria*-specific universal primers 27f and 1492r (70). Pairwise evolutionary distances between SMMA_5 and other *Paracoccus* species were determined based on their aligned 16S rRNA gene sequences. After this, a neighbor-joining tree was constructed following the majority rule, and strict consensus out of 1,000 phylogenetic trees, using the software package MEGAX (71). To check the robustness of the tree's branching pattern, bootstrap values were calculated based on 1,000 replicates.

After the *de novo* sequencing and assembly of their genomes (methods given below), the phylogeny of SMMA_5 and SMMA_5_TC was reconstructed based on 92 conserved marker genes, in relation to the other species of *Paracoccus* for which whole-genome sequences were available in the GenBank database (a total of 44 nonredundant genomes for different *Paracoccus* species were included in the analysis on a "one genome per species" basis; two more genomes belonging to *Paracoccus* species were also there in the database but they were not considered due to their low levels of completeness; Table S1). Majority of the genes considered for phylogeny reconstruction (i.e., 67 out of 92), belonged to that functional (metabolic) category of COGs which involves translation, including ribosome structure and biogenesis. The up-to-date bacterial core gene (UBCG) set was used to identify and align these marker genes (72). Final tree construction from the aligned marker genes was created using RAxML version 8 (73). Final visualization of the resultant tree was done using interactive Tree of Life (iTOL) version 4 (74).

**Growth experiments.** Growth or decline in the CFU count of *Paracoccus* cultures at different temperatures was first tested in MST medium (pH 7.0) containing 20 mM thiosulfate as the sole source of energy and electrons (75). Generation time (g) was determined by putting the values of initial (0 h) and final (12 h) CFU count $mL^{-1}$ culture to the equation given below (76), where t denotes the time of incubation, N stands for the CFU count recorded after the time t, and $N_0$ denotes the CFU count recorded at the 0 h of incubation.

$$g = \frac{\log 2 \, (t - t_0)}{\log N - \log N_0}$$

Subsequently, SMMA_5 and SMMA_5_TC were tested at different temperatures in Luria broth, and then in different concentrations of R2A (diluted up to 1,000 times, thereby affording an extremely low absolute concentration of organic carbon) that in turn were adjusted to different levels of pH (7.0 to 9.0). Corresponding experiments for *E. coli* K-12 were carried out in MS supplemented with 4 g $L^{-1}$ dextrose (MSD, pH 7.0 [75]). Growth or decline in the CFU count of SMMA_5 and SMMA_5_TC at different temperatures was further tested in MST supplemented with 4 mM $Na_2B_4O_7.10H_2O$ (i.e., 16 mM B), 1 mM $LiOH.H_2O$ (i.e., 1 mM Li), and/or 10 mM glycine-betaine or N,N,N-trimethylglycine ($C_5H_{11}NO_2$); the three solute-fortified variants of MST were referred to as MSTB, MSTL, and MSTG, respectively. Notably, the boron and lithium concentrations used in the culture media were close to those recorded previously in the habitat of SMMA_5 (7, 14, 23).

To test the growth or decline in the CFU count of a given strain under a particular condition, a seed culture was first prepared in the strain's designated maintenance medium via incubation at 37°C. Inoculum from the log phase seed culture was transferred to the experimental broth, and incubated at the temperature specified for the experiment. To determine the number of CFUs present $mL^{-1}$ of an experimental culture at a given time point of incubation (including the 0 h), its various dilution grades (created using 0.9% [wt/vol] NaCl) were plated in triplicates on to R2A agar (for SMMA_5 and SMMA_5_TC) or Luria agar (for LMG 4218 and K-12), and single colonies were counted in each of them after 36 h of incubation at 37°C. Colony counts in the different dilution plates were multiplied by their respective dilution factors, then summed up across the plates and dilution grades, and finally averaged to get the number of CFUs that were present $mL^{-1}$ of the experimental culture. Cellular growth yield after a given time period of incubation was expressed as what percentage of the 0 h CFU count was present in the culture after that period of incubation.

**Testing the effect of thermal conditioning cycles on CFU retention.** SMMA_5 was subjected to one or five thermal conditioning cycles (Fig. 4B), both of which started with 1% inoculum transfer to a fresh R2A broth, but subsequently involved one or five iteration(s) of the following incubation cycle, respectively: 12 h at 50°C and then 12 h at 37°C. In the case of five iterations, two consecutive incubation cycles were separated by an interval of 48 h at 25°C. In either case (whether thermal-conditioning cycles were one or five), cells from the last 37°C-grown culture were dilution-plated on R2A agar, and six discrete colonies were selected as six individual sibling strains of SMMA_5. Finally, each member of the two strain sets, designated C1_S1 to C1_S6 (Fig. 5A, B and E and F) and C5_S1 to C5_S6 (Fig. 5C, D and G and H) according as they originated from one or five conditioning cycles, was tested individually for its CFU growth/decline at different temperatures following the procedure already described above.

**Determining the percentage of viable cells in a culture.** CFU count gave the estimate of what percentage of cells in a culture, at a given time point of incubation, retained the ability to divide readily, i.e., to form colonies in R2A/Luria agar plates after 36 h at 37°C. In contrast, what percentage of the cell population remained viable or metabolically active was tested by checking the cells' ability to accumulate the nontoxic and nonfluorescent molecule fluorescein diacetate and hydrolyze it to fluorescein, which in turn was detected via flow cytometry (77). Post incubation, cells were precipitated from a 100 mL volume of the test culture by centrifugation at 6,000 *g* for 20 min at 4°C. The supernatant was discarded, and the cell pellet resuspended in 2 mL 0.9% NaCl solution. Then, 4 $\mu$L of FDA (Sigma, USA) solution (5 mg FDA $mL^{-1}$ dimethyl sulfoxide) was added to the resuspended solution and the cells were incubated for 15 to 20 min at 37°C. Incubated cells were washed and resuspended again in 0.9% NaCl solution (final resuspension was done in 500 $\mu$L), and eventually analyzed using a FACSVerse flow cytometer (Becton, Dickinson, USA). Fluorescence was measured through the excitation wavelengths 475 to 495 nm and the emission wavelengths 520 to 530 nm. A total of 10,000 randomly taken cells were analyzed for each sample and dot plots were generated using the fluorescence level of each cell as a function of its forward scattering of a 488 nm wave detected by a photodiode array detector. The data were analyzed using the software BD FACSuite (Becton, Dickinson) by specific quadrant gating for each experiment that in turn were predetermined based on unstained samples. In all the individual gating experiments, <2% cells within the FDA-less controls exhibited fluorescence; this was the maximum proportion at which false positives could be there within the different values reported for viable cell percentage.

**Determining glycine-betaine concentration in spent media.** After incubation in MST for 12 h at 50°C, SMMA_5 or SMMA_5_TC cells were pelleted out from 100 mL spent medium by centrifugation at 6,000 *g* for 20 min. The cell-free supernatant was lyophilized down to 1 mL and then filtered by passing through a 0.22-$\mu$m cellulose acetate filter (Sartorius Stedim Biotech, Germany). Next, 25 $\mu$L of the filtrate was analyzed by high performance liquid chromatography (HPLC), followed by standard UV detection (78), using a Waters platform that encompassed a FlexInject sample-injector, a 1525 Binary HPLC pump, a 2998 photodiode array detector, and the software Breeze 2.0 (Waters Corporation, USA). Isocratic elution was carried out in a Hypersil SCX column having a 5-$\mu$m particle size, 250-mm length, and 4.6-mm inner diameter (Phenomenex, USA), using a mixture of disodium phosphate buffer (0.05 M, pH 4.7) and methanol (95:5, vol/vol) as the mobile phase (injection volume: 25 $\mu$L; flow rate: 1 mL $minute^{-1}$ at 25°C; detection wavelength: 195 nm). Zero, 2, 4, 8, and 10 mM glycine-betaine standards (Sigma, USA) were used to generate the calibration curve. Glycine-betaine in the sample was identified by comparing its retention time with that of the standard; purity of chromatographic peaks was checked using Breeze 2.0. Quantification was based on the area of the chromatographic peak using the calibration

curve Y = mX + C, where Y is the peak-area, X is the concentration of glycine-betaine, m is the slope of the calibration curve, and C is the curve intercept for the sample.

**Analysis of genomes and putative proteomes.** The whole-genome shotgun sequences of SMMA_5 and SMMA_5_TC were determined using an Ion S5 system (Thermo Fisher Scientific, USA) as described previously (79, 80). The assembled genome sequences were annotated for potential ORFs or CDSs using the Prokaryotic Genome Annotation Pipeline of the National Center for Biotechnology Information, USA. Completeness of the new genomes, or that of the comparator *Paracoccus* species, was determined using CheckM 1.0.12 based on what percentage of the genus-specific marker genes was present in the genome considered (81). Pairwise DNA-DNA hybridization experiments were carried out using the online software Genome-to-Genome Distance Calculator 3.0 (82). Orthologous gene-based average nucleotide identity between genome pairs was calculated using ChunLab's online ANI Calculator (83).

To identify potential modifications in the genome of SMMA_5_TC with respect to the SMMA_5 genome, the two sequences were first aligned using the software MAFFT (84) with default parameters. The software SNP-sites (85) was then used to identify single nucleotide polymorphisms (SNPs) from the alignment; in order to enumerate the different transition and transversion events, the variant call format (VCF) file generated was processed with the help of VCFtools (86), while the software SnpEff (87) was used to annotate all the SNPs at the nucleotide as well as translated amino acid sequence levels.

The core, and species-specific unique, genes of *Paracoccus* members, including SMMA_5, were delineated with the help of the bacterial pan genome analysis pipeline BPGA version 1.3.0 (88) as described previously (14). This analysis involved the SMMA_5 genome plus the 44 nonredundant genome sequences of *Paracoccus* species (each having a completeness level of ≥97%) available in GenBank (Table S1). In the working data set, one genome sequence was included per species, preferably that of the type strain, unless the same was unavailable in GenBank. SMMA_5_TC was left out because two sibling genomes with identical gene-contents would have distorted the results pertaining to the identification of unique genes.

Putative protein sequence catalogs were derived from the genomes of SMMA_5, SMMA_5_TC, and the 44 comparator species of *Paracoccus* which have nearly complete genome sequences available in GenBank. Subsequently, the pI profile of each catalog was determined using the software Isoelectric Point Calculator that utilizes the Henderson-Hasselbach equation and calculates pKa values for all the individual protein sequences present in a catalog (89).

For SMMA_5 and 14 other *Paracoccus* species occupying the same clade in the phylogenomic tree (Fig. 1), complete gene catalogs were downloaded from GenBank. Within each catalog, genes attributable to COGs (90) were first selected, and then COG counts under different functional (metabolic) categories were determined. Whether COG count under a functional category was significantly high or low in a particular genome was determined by Chi-Square test, as described previously (14), using the contingency chart shown in Table S11 (a *P* value less than 0.001 was considered as the cutoff for inferring whether the presence of COGs under a category was high or low for a species). Hierarchical clustering was carried out to quantitatively decipher the relatedness between the genomes in terms of their enrichment of various COG categories (91). The heat map illustrating the results of hierarchical cluster analysis was constructed with the help of an R script using complete linkage method (14).

**Data availability.** The PCR-amplified 16S rRNA gene sequence of SMMA_5 was deposited to GenBank under the accession number LN869532. The assembled whole-genome sequences of SMMA_5 and SMMA_5_TC were deposited to GenBank with the accession numbers WIAB00000000 and WIAC00000000, respectively. The two raw read data sets were deposited to the Sequence Read Archive of the National Center for Biotechnology Information, USA, under the BioProject number PRJNA296849, with the run accession numbers SRR10281900 and SRR10281899, respectively.

## SUPPLEMENTAL MATERIAL

Supplemental material is available online only.
**SUPPLEMENTAL FILE 1**, PDF file, 1.8 MB.
**SUPPLEMENTAL FILE 2**, XLSX file, 0.1 MB.

## ACKNOWLEDGMENTS

This research was financed by Bose Institute (via intramural faculty grants) as well as the Science and Engineering Research Board (SERB), Government of India (GoI) (SERB grant number EMR/2016/002703). N.M. received fellowships from SERB and Bose Institute. C.R. received a fellowship from the University Grants Commission, GoI. S.C. received a fellowship from the Department of Biotechnology, GoI. J.S. and S.D. obtained their fellowships from Council of Scientific and Industrial Research, GoI. S.B. received a fellowship from Bose Institute.

W.G. conceived the research program, designed the experiments, interpreted the results, and wrote the paper. N.M. planned and performed the experiments, harnessed all the modules of the study, and analyzed and curated the data. C.R. led the studies of genomics and carried out experiments of physiology. S.C., S.D., and S.B. carried out the physiological studies, while J.S. performed studies of genomics. R.C. made critical

intellectual contributions to the overall work as well as the paper. All the authors read and vetted the paper.

We have no conflict of interest to declare.

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
