## [Reviewer comments · Microbiology Spectrum]

Microbiology Spectrum

Thermal endurance by a hot-spring-dwelling phylogenetic relative of the mesophilic *Paracoccus*

Nibendu Mondal, Chayan Roy, Sumit Chatterjee, Jagannath Sarkar, Subhajit Dutta, Sabyasachi Bhattacharya, Ranadhir Chakraborty, and Wriddhiman Ghosh

Corresponding Author(s): Wriddhiman Ghosh, Bose Institute

Review Timeline:

Submission Date:	May 6, 2022
Editorial Decision:	July 27, 2022
Revision Received:	September 21, 2022
Accepted:	September 24, 2022

Editor: Jeffrey Gralnick

Reviewer(s): The reviewers have opted to remain anonymous.

Transaction Report:

DOI: <https://doi.org/10.1128/spectrum.01606-22>

July 27, 2022

Dr. Wriddhiman Ghosh
Bose Institute
Microbiology
P-1/12, CIT Road Scheme VIIM
Kolkata 700 054 WB India
Kolkata, West Bengal 700054
India

Re: Spectrum01606-22 (Thermal endurance by a hot-spring-dwelling phylogenetic relative of the mesophilic *Paracoccus*)

Dear Dr. Wriddhiman Ghosh:

Thank you for submitting your manuscript to Microbiology Spectrum. Your manuscript was reviewed by an expert in the field and myself (we had another reviewer withdraw due to personal issues). While I think that having some more refined growth data would be nice (as suggested by Reviewer 2), I will leave this up to you. The manuscript is written quite well and is careful not to confuse growth with survival / endurance. When submitting the revised version of your paper, please provide (1) point-by-point responses to the issues raised by the reviewer (and myself) as file type "Response to Reviewers," not in your cover letter, and (2) a PDF file that indicates the changes from the original submission (by highlighting or underlining the changes) as file type "Marked Up Manuscript - For Review Only". Please use this link to submit your revised manuscript - we strongly recommend that you submit your paper within the next 60 days or reach out to me. Detailed instructions on submitting your revised paper are below.

Link Not Available

Sincerely,

Jeffrey Gralnick

Senior Editor, Microbiology Spectrum

Journals Department
Editor comments:

The manuscript by Mondal et al. seeks to understand and explore thermal endurance of a *Paracoccus* strain (SMMA_5) isolated from a hot spring. The authors evolve a variant of SMMA_5 through a regime of high-temperature (50C) growth conditions, yielding a strain denoted as SMMA_5_TC. The authors have done a large amount of work to determine thermal endurance characteristics of these strains, with comparisons to a mesophilic *Paracoccus* isolate and a lab strain of *E. coli*. Partial genomes were also generated for both strains and some differences noted by the authors. The authors describe the differences between the strains appropriately as potential mechanisms for enhancing survival at 50C, but these hypotheses were not experimentally validated. While there are some additional experiments that would help the authors tell a clearer story (growth vs survival at 50C

/ complete genomes), there is a lot of excellent data presented and I think researchers who think about thermal adaptation will be interested in the work.

There are a couple references to memory / memorize when the authors actually mean evolution / adaptation / genetic change (e.g. lines 41 and 365). I don't think this anthropomorphism is particularly useful.

It is unclear from the data presented in Figure 2 if SMMA_5_TC is actually capable of growth at 50C or if the strain survives this condition better than the parent strain (SMMA_5). The selection regime shown in Figure 4A doesn't necessarily require that the strain grow at 50C, but rather survive for 12 hours prior to shifting down to 37C. The authors are clear in the manuscript and do not outright claim to have shown growth. A growth experiment could clear this up, but I don't think is required.

Comparisons of these strains is hampered by the incomplete genomes generated by short Illumina reads. For future work the authors should consider coupling some low-cost long-read sequencing (such as Nanopore) to combine with their short-read sequences to generate finished genomes.

Specific comments for consideration.

Line 146 - perhaps mention where LMG 4218 was isolated from for comparison purposes?

Line 188 - please rephrase this. I think I know what you mean, but the heading is awkward.

Line 208 - is this release due to cell lysis?

Line 505 - this section is quite speculative and does not add meaningfully to the manuscript, in my opinion.

Line 531 - does addition of Li lead to 'metabolic deceleration' as well? Might this account for the enhanced thermal tolerance when the medium was amended with Li?

Line 566 - was the SMMA_5_TC variant also deposited in the culture collection?

Figure 2 legend: please define MST and MSD here so readers do not have to refer back to the methods.

Line 617 - please clarify the medium names here as 4 conditions are given but only 3 names are provided.

Reviewer comments:

Reviewer #2 (Comments for the Author):

Title: Thermal endurance by a hot-spring-dwelling phylogenetic relative of the mesophilic *Paracoccus*

Summary: A *Paracoccus* strain isolated from high-temperature environments was studied. Cell viability was examined after incubation at varying temperatures. The genome was sequenced.

Comments: The authors may have isolated an interesting member of the genus *Paracoccus*. There are cases in which phylogenetically related organisms display significantly different growth temperatures, and this study may demonstrate such a case in the *Paracoccus* genus. However, some basic biochemical properties are missing, and should be provided. Although the authors focus on temperature endurance, a growth temperature analysis at varying temperatures is essential to understand the newly isolated strain and interpret the results of experiments on temperature endurance.

Specific comments

1. Cells of the newly isolated strain(s), along with a conventional *Paracoccus* strain, should be grown at 37, 40, 45, 50, 55, 60, 65 degrees in liquid culture. The specific growth rates must be calculated and compared.
2. Searching for links between metabolism and thermal endurance may be interesting, but the metabolism should be described in a more specific manner. The authors should distinguish between catabolism and anabolism for metabolism such as for amino acids, and compare the features with conventional *Paracoccus* strains with genome sequences. The connection to increased/decreased thermal endurance should be explained (eg. Some amino acids might act as compatible solutes. Are their biosynthesis pathways strengthened?).
3. The number and types of heat shock proteins and the chaperone proteins should be compared with those from conventional *Paracoccus* strains. There may be an increase compared to conventional *Paracoccus* strains.
4. As suggested in the discussion, the authors should compare the number and types of ion-channels in SMMA_5 with those from conventional *Paracoccus* strains. The authors could also include ABC transporters.
5. There seems to be discussion on aspects of cell growth, such as DNA replication, with temperature in the discussion. This makes the growth experiments (specific growth rate analyses) indicated in comment 1 all the more important.

Staff Comments:

Preparing Revision Guidelines

Please return the manuscript within 60 days; if you cannot complete the modification within this time period, please contact me. If you do not wish to modify the manuscript and prefer to submit it to another journal, please notify me of your decision immediately so that the manuscript may be formally withdrawn from consideration by Microbiology Spectrum.

Response to the comments of Editor / Reviewer #1

Comment 1: The manuscript by Mondal et al. seeks to understand and explore thermal endurance of a *Paracoccus* strain (SMMA_5) isolated from a hot spring. The authors evolve a variant of SMMA_5 through a regime of high-temperature (50C) growth conditions, yielding a strain denoted as SMMA_5_TC. The authors have done a large amount of work to determine thermal endurance characteristics of these strains, with comparisons to a mesophilic *Paracoccus* isolate and a lab strain of *E. coli*. Partial genomes were also generated for both strains and some differences noted by the authors. The authors describe the differences between the strains appropriately as potential mechanisms for enhancing survival at 50C, but these hypotheses were not experimentally validated. While there are some additional experiments that would help the authors tell a clearer story (growth vs survival at 50C complete genomes), there is a lot of excellent data presented and I think researchers who think about thermal adaptation will be interested in the work.

RESPONSE: We thank you for these comments, and believe you appreciated the underlying science of this study. We also fully agree with your concerns, so have now dealt with each one of them in course of this revision.

Comment 2: There are a couple references to memory memorize when the authors actually mean evolution adaptation genetic change (e.g. lines 41 and 365). I don't think this anthropomorphism is particularly useful.

RESPONSE: We agree that the current paper does not have any direct data to substantiate such remarks in the context of the evolution / adaptation of the hot spring *Paracoccus*, so we have now removed the words memory, memorize, etc. from wherever it had been used previously with reference to SMMA_5 or SMMA_5_TC (*please see the changes made in lines 47, 414, 444 and 673 of the Track Changes file*).

However, in the discussion, we have retained the word memory for those comparator cases (*lines 422 and 425 of the Track Changes file*) where similar phenomena reported for other organisms have been referred to as bacterial memory in the concerned papers. Notably, most of the papers cited for those comparator instances (*references numbered as 33, 34, 36, and 37 in the revised text*) have used the word memory in the title itself; for instance:

33. Casadesús J, D'Ari R. 2002. Memory in bacteria and phage. *BioEssays* 24:512–518. <https://doi.org/10.1002/bies.10102>

34. Govers SK, Mortier J, Adam A, Aertsen A. 2018. Protein aggregates encode epigenetic memory of stressful encounters in individual *Escherichia coli* cells. *PLoS Biol* 16:e2003853. <https://doi.org/10.1371/journal.pbio.2003853>

36. Lee CK, Anda J de, Baker AE, Bennett RR, Luo Y, Lee EY, Keefe JA, Helali JS, Ma J, Zhao K, Golestanian R, O'Toole GA, Wong GCL. 2018. Multigenerational

memory and adaptive adhesion in early bacterial biofilm communities. *Proc Natl Acad Sci U S A* 115:4471–4476. <https://doi.org/10.1073/pnas.17200711115>

37. Yang CY, Bialecka-Fornal M, Weatherwax C, Larkin JW, Prindle A, Liu J, Garcia-Ojalvo J, Süel GM. 2020. Encoding membrane-potential-based memory within a microbial community. *Cell Syst* 10:417–423. <https://doi.org/10.1016/j.cels.2020.04.002>

Comment 3: It is unclear from the data presented in Figure 2 if SMMA_5_TC is actually capable of growth at 50C or if the strain survives this condition better than the parent strain (SMMA_5). The selection regime shown in Figure 4A doesn't necessarily require that the strain grow at 50C, but rather survive for 12 hours prior to shifting down to 37C. The authors are clear in the manuscript and do not outright claim to have shown growth. A growth experiment could clear this up, but I don't think is required.

RESPONSE: We thank you for appreciating the clarity of our data presentation details.

We also agree with your point that a growth experiment could resolve the actual extent of cell division in SMMA_5_TC at 50°C.

But, as you have aptly noted at the end of your comment, such data are better kept out from the purview of the current manuscript because the paper is already loaded with huge data sets of physiology, and once we present a full-fledged growth curve (kinetic data) for SMMA_5_TC, to put things into their right perspective, we shall have to present corresponding data for SMMA_5 as well as all the comparator strains (that too under the different media and temperature conditions which are already in question).

Having said that, we are indeed happy to convey that the extensive (multi-conditional) transcriptome analyses, which we are already carrying out for SMMA_5 and SMMA_5_TC, begin with in-depth growth experiments under different media and temperature conditions.

Comment 4: Comparisons of these strains is hampered by the incomplete genomes generated by short Illumina reads. For future work the authors should consider coupling some low-cost long-read sequencing (such as Nanopore) to combine with their short-read sequences to generate finished genomes.

RESPONSE: We agree, and have already progressed considerably on the Phase II study involving genome completion (indeed by combining long ONT reads with short Illumina reads), followed by extensive transcriptome analysis for SMMA_5 and SMMA_5_TC (we earnestly plan to communicate that paper to this journal very soon).

So far as the present paper is concerned, the two, approximately 98.5% complete, shotgun genomes are sufficiently reliable for the limited molecular biological

conclusions that have been currently drawn (off the records, the ONT-completed genomes added ~30 kb sequence, i.e. only 20-25 genes, to the existing genomes).

Specific comments

Comment 1 Line 146: perhaps mention where LMG 4218 was isolated from for comparison purposes

RESPONSE: We agree, and have now added the required information along with the relevant reference (Robertson, L.A. and Kuenen, J.G., 1983. *Thiosphaera pantotropha* gen. nov. sp. nov., a facultatively anaerobic, facultatively autotrophic sulphur bacterium. *Microbiology*, 129(9), pp.2847-2855). Please see lines 160-161 of the Track Changes file.

Furthermore, to maintain uniformity, we have also added similar information for the other comparator strain (*Escherichia coli* K-12) used in this study. Please see lines 168-169 of the Track Changes file.

Comment 2 Line 188: please rephrase this. I think I know what you mean, but the heading is awkward.

RESPONSE: We agree, so have now fixed the problem by removing the subjective words “cell viability” and “divisibility” from the heading, and replacing them by the data-centric terminologies “proportion of metabolically active cells” and “proportion of CFUs” (please see lines 207-208 of the Track Changes file).

Comment 3 Line 208: is this release due to cell lysis

RESPONSE: The data presented in the Results sections just above this line showed that the cells of both SMMA_5 and SMMA_5_TC remain in a metabolically active, but non-dividing or hardly dividing, state after 12 h incubation in MST at 50°C. Under this condition, therefore, the cells are healthy, and capable of synthesizing and transporting essential biomolecules. In order to provide a subtle clarification on this issue we have now added a small clause to the existing sentence on glycine-betaine release by SMMA_5 and SMMA_5_TC to the spent medium (please see lines 228-230 of the Track Changes file).

Comment 4 Line 505: this section is quite speculative and does not add meaningfully to the manuscript, in my opinion.

RESPONSE: We agree that the discussion pertaining to the predicted pI values of all putative proteins of the different *Paracoccus* species (including SMMA_5) is largely theoretical. We have now edited the text to contextualize the issue in a more straight forward and logical way (please see lines 612-616 and 624-643 of the Track Changes file).

As for the relevance and necessity of this section, the following points are noteworthy.

It has been shown previously that at equivalent temperature levels, hot spring waters having neutral to moderately alkaline pH generally harbor greater microbial diversities than their acidic counterparts. In view of this, and also in consideration of the circum-neutral pH (diurnal range: 7.2-8.0) of the Lotus Pond vent-water (the habitat of the test organism), we investigated the effects of pH 7.0-9.0 on the high temperature growth/survival of SMMA_5. *This point was missing from the previous manuscript, so we have now added this justification to the Introduction (please see lines 113-117 of the Track Changes file).*

Again, from a molecular perspective, pH of the chemical milieu is a key biophysical parameter for protein stability, and thereby overall microbial adaptation to different temperature regimes. Consequently, it was essential for the present study to investigate the pH optima for thermal endurance by the hot spring *Paracoccus*: we did so and found that pH 7.5 was best for growth at 37°C and CFU-retention at 50°C and 70°C. As an aftermath of these data, the biophysical discussions centered on pI values of proteins became necessary. *In the previous manuscript, this contextual explanation was missing from the section of the Discussion in question; we have now added it (please see lines 612-616 of the Track Changes file).*

Additionally, during the current revision, this Discussion section on biophysical adaptations also turned out to be very useful in accommodating the fresh critical discourse (*please see lines 581-611 of the Track Changes file*) emanating from the new analyses pertaining to amino acid frequencies in *Paracoccus* proteomes, which in turn were warranted by Reviewer 2.

Comment 5 Line 531: does addition of Li lead to 'metabolic deceleration' as well Might this account for the enhanced thermal tolerance when the medium was amended with Li

RESPONSE: We agree with your postulation, but the data available at this point of time does not allow us to be very specific about lithium; or for that matter any of three solutes which enhanced thermal tolerance.

Based on the results in hand, only this much can be inferred safely: metabolic deceleration is intrinsic to the SMMA_5 strategy for thermal endurance, and the environmental solutes plausibly augment that intrinsic metabolic plan.

The whole issue was discussed in the section starting at line 531 of the previous manuscript (*now, line number 645 of Track Changes file*).

Comment 6 Line 566: was the SMMA_5_TC variant also deposited in the culture collection

RESPONSE: No, SMMA_5_TC has not yet been deposited to any culture collection center (CCC). The main issue here is that CCCs generally maintain strains via lyophilization and long-term storage, with viability checks and renewals of the biomass at long intervals. But to maintain the characteristic features of the strain SMMA_5_TC the culture needs to be transferred every 14 days (via incubation at 50°C for 12 h and then at 37°C for another 12 h) and stored as a running culture at 25°C; we are in talks with a few CCCs to figure out whether any one of them agrees to undertake this tedious transfer regime.

Comment 7 Figure 2 legend: please define MST and MSD here so readers do not have to refer back to the methods.

RESPONSE: We agree, so have now added the full-forms of MST and MSD in the legend of Figure 2.

Comment 8 Line 617: please clarify the medium names here as 4 conditions are given but only 3 names are provided.

RESPONSE: Indeed media types are four – **MST**, **MSTB**, **MSTL** and **MSTG**, and we agree that the previous articulation was a tad confusing, so have now made a small edit to make the point clear (please see the mark-up below).

Under this Methods section, the first line (numbered as 732 in the Track Changes file) reads

“Growth or decline in the CFU-count of *Paracoccus* cultures at different temperatures was first tested in **MST** medium (pH 7.0) containing 20 mM thiosulfate as the sole source of energy and electrons (75)”.

Line number 745-751 (of the Track Changes file) goes on to add “Growth or decline in the CFU-count of SMMA_5 and SMMA_5_TC at different temperatures was further tested in **MST** supplemented with 4 mM Na₂B₄O₇·10H₂O (i.e., 16 mM B), 1 mM LiOH·H₂O (i.e., 1 mM Li) and/or 10 mM glycine-betaine or N,N,N trimethylglycine (C₅H₁₁NO₂); the three solute-fortified variants of **MST** were referred to as **MSTB**, **MSTL** and **MSTG** respectively.”

Response to the comments of Reviewer #2

Title Thermal endurance by a hot-spring-dwelling phylogenetic relative of the mesophilic *Paracoccus*

Summary A *Paracoccus* strain isolated from high-temperature environments was studied. Cell viability was examined after incubation at varying temperatures. The genome was sequenced.

Comments The authors may have isolated an interesting member of the genus *Paracoccus*. There are cases in which phylogenetically related organisms display significantly different growth temperatures, and this study may demonstrate such a case in the *Paracoccus* genus. However, some basic biochemical properties are missing, and should be provided. Although the authors focus on temperature endurance, a growth temperature analysis at varying temperatures is essential to understand the newly isolated strain and interpret the results of experiments on temperature endurance.

RESPONSE: We thank the Reviewer for appreciating the phenomenology underlying the present study, and helping improve the manuscript technically and in terms of interpretations. We have taken positive actions with respect to all your suggestions and each of them are narrated below.

As warranted by you, we have now carried out a growth analysis at varying temperatures for the newly isolated strain SMMA_5 alongside the conventional *Paracoccus* strain LMG 4218 and *E. coli* K-12. We also valued your suggestion pertaining to the delineation of basic biochemical properties for the new isolate, as that would help us know the new strain better in the context of its close phylogenetic relatives. However, we just stopped short of presenting a comparative dataset of biochemical attributes in this paper because such a dataset (which holds major implications for the exact taxonomic identification of the strain) might just shift the focus aside from the thermal endurance aspect which is central to the current narrative (we would surely present these information in a subsequent paper of taxonomy in *Int J Syst Evol Microbiol*).

Specific comments

Comment 1: Cells of the newly isolated strain(s), along with a conventional *Paracoccus* strain, should be grown at 37, 40, 45, 50, 55, 60, 65 degrees in liquid culture. The specific growth rates must be calculated and compared.

RESPONSE: We agree: the organisms have now been grown at 37°C, 40°C and 45°C, and their specific growth rates calculated at these temperatures. The newly incorporated data, discussions and methods which pertain to this aspect can be seen in lines 154-173, 646-653 and 734-739 of the Track Changes file.

A comparison between the generation times of SMMA_5 and *P. pantotrophus* LMG 4218, recorded at 37°C, 40°C and 45°C (in MST medium), indeed showed that with

increasing temperature growth decelerated in both the strains, but the slowdown was less in SMMA_5, as compared to LMG 4218. Albeit no growth was observed for either strain at 50°C under the routine culture conditions, a small increase in CFU-count, reflective of extremely slow growth rate, was observed for the hot spring *Paracoccus* alone after certain habitat-inspired manipulations were rendered in the strain's maintenance history and/or the chemical composition of the culture media.

Comment 2: Searching for links between metabolism and thermal endurance may be interesting, but the metabolism should be described in a more specific manner. The authors should distinguish between catabolism and anabolism for metabolism such as for amino acids, and compare the features with conventional *Paracoccus* strains with genome sequences. The connection to increased decreased thermal endurance should be explained (eg. Some amino acids might act as compatible solutes. Are their biosynthesis pathways strengthened).

RESPONSE: We agree, and have now added the precise identities the unique SMMA_5 genes for the enriched COG categories (i) amino acid transport and metabolism, (ii) cell wall/membrane/envelope biogenesis, and (iii) lipid transport and metabolism, in relation to

- all 44 *Paracoccus* species having near-complete genome sequences available, as well as
- the 14 phylogenomically closest relatives.

The newly incorporated data and discussions which pertain to this aspect can be seen in lines 338-347 and 539-546 of the Track Changes file, alongside the new Table S5.

Furthermore, we have also analyzed the relative abundances of the 20 individual amino acids (expressed as the percentage of all amino acids present) across the putative (*in silico* translated) proteomes of SMMA_5 and its 14 closest phylogenomic relatives (new data and discussions pertaining to this aspect can be seen in lines 347-361 and 581-611 of the Track Changes file, alongside the new Table S10.)

Comment 3: The number and types of heat shock proteins and the chaperone proteins should be compared with those from conventional *Paracoccus* strains. There may be an increase compared to conventional *Paracoccus* strains.

RESPONSE: We agree, and have now worked more on this. A comparison of SMMA_5 with a few of its phylogenomically closest relatives, in relation to the types and numbers of their genes for heat shock proteins and chaperones, was already there in the previous manuscript (previous Table S5); in the revised manuscript we have expanded that analysis by including all the 14 *Paracoccus* species that occupied the same clade as SMMA_5 in the phylogenomic tree. The updated comprehensive data can be found in the new Table S6, while corresponding changes to the text can be seen in lines 374-382 and 537-546 of the Track Changes file.

Comment 4: As suggested in the discussion, the authors should compare the number and types of ion-channels in SMMA_5 with those from conventional *Paracoccus* strains. The authors could also include ABC transporters.

RESPONSE: We agree, and have now compared the number and types of genes for channel proteins and porins / aquaporins; ABC transport systems; cobalt, copper, magnesium and/or nickel transporters; uni- / sym- / antiporters; ECF class transporters; Ton and Tol transporters; and TRAP transporters

in SMMA_5 with those present in the 14 phylogenomically closest *Paracoccus* species.

The new data and discussions pertaining to this aspect can be seen in *lines 428-437 of the Track Changes file, alongside the new Table S9.*

Comment 5: There seems to be discussion on aspects of cell growth, such as DNA replication, with temperature in the discussion. This makes the growth experiments (specific growth rate analyses) indicated in comment 1 all the more important.

RESPONSE: We agree, so have now grown SMMA_5 alongside the conventional *Paracoccus* strain LMG 4218 (and also *E. coli* strain K-12) at different temperatures and calculated their specific growth rates at those temperatures. As already mentioned in our response to your Comment 1, these new experiments, data and discussions (*incorporated in lines 154-173, 646-653 and 734-739 of the Track Changes file*) have indeed substantiated our inferences on metabolic deceleration as central to thermal endurance.

September 24, 2022

Dr. Wriddhiman Ghosh
Bose Institute
Microbiology
P-1/12, CIT Road Scheme VIIM
Kolkata 700 054 WB India
Kolkata, West Bengal 700054
India

Re: Spectrum01606-22R1 (Thermal endurance by a hot-spring-dwelling phylogenetic relative of the mesophilic *Paracoccus*)

Dear Dr. Wriddhiman Ghosh:

Your manuscript has been accepted, and I am forwarding it to the ASM Journals Department for publication. You will be notified when your proofs are ready to be viewed.

Sincerely,

Jeffrey Gralnick
Editor, Microbiology Spectrum

Journals Department
2: Accept

1: Accept